# Open-Vocabulary Part Segmentation via Progressive and Boundary-Aware Strategy

**Xinlong Li[1]\*   Di Lin[1]\*   Shaoyiyi Gao[1]   Jiaxin Li[2]   Ruonan Liu[3]   Qing Guo[4]†**

[1]Tianjin University, China    [2]Southwest University, China
[3]Shanghai Jiao Tong University, China    [4]VCIP, CS, Nankai University, China
`lxl_zs@tju.edu.cn`

## Abstract

Open-vocabulary part segmentation (OVPS) struggles with structurally connected boundaries due to the inherent conflict between continuous image features and discrete classification mechanism. To address this, we propose PBAPS, a novel training-free framework specifically designed for OVPS. PBAPS leverages structural knowledge of object-part relationships to guide a progressive segmentation from objects to fine-grained parts. To further improve accuracy at challenging boundaries, we introduce a Boundary-Aware Refinement (BAR) module that identifies ambiguous boundary regions by quantifying classification uncertainty, enhances the discriminative features of these ambiguous regions using high-confidence context, and adaptively refines part prototypes to better align with the specific image. Experiments on Pascal-Part-116, ADE20K-Part-234, PartImageNet demonstrate that PBAPS significantly outperforms state-of-the-art methods, achieving 46.35% mIoU and 34.46% bIoU on Pascal-Part-116. Our code is available at https://github.com/TJU-IDVLab/PBAPS.

## 1   Introduction

Semantic segmentation aims to assign each pixel in an image to a predefined class. Traditional methods [1, 2, 3, 4, 5, 6] are based on supervised learning with labeled data [7, 8, 9] and have achieved significant progress in closed-set but exhibit limited zero-shot generalizability in open-world scenarios. Open-vocabulary semantic segmentation (OVSS) [10, 11, 12] addresses this limitation by leveraging pre-trained vision-language models (VLM) [13, 14, 15] to enable segmentation of unseen classes. Existing OVSS methods can be broadly categorized into two paradigms: (1) direct matching via cross-modal similarity [16, 12, 17] and (2) mask classification frameworks [18, 19, 20] based on feature clustering. Although effective for object-level segmentation, their part-level performance degrades significantly [21]. Open-vocabulary part segmentation (OVPS) [22, 17, 23] faces the additional challenge of ambiguous part boundaries.

We subdivide boundaries into three types: object boundary, structurally connected part boundary, and non-structurally connected part boundary. Structurally connected parts refer to anatomically adjacent components with direct physical connections (e.g., cat head and neck). Existing OVPS methods, including multigranularity segmentation based on object-part modeling [24, 25, 26] and fine-grained feature enhancement approaches [27, 28], commonly exhibit inaccuracies in segmenting intra-object structurally connected part boundaries (Figure 1). These approaches neglect the key distinction between OVSS and OVPS: object-level segmentation depends on distinct feature variations across boundaries, while part-level segmentation encounters smooth and continuous feature transitions (e.g.,

---

\*Co-first authors.

†Corresponding author.

39th Conference on Neural Information Processing Systems (NeurIPS 2025).

fur texture/color changes) at structurally connected boundaries, in contrast to the moderate local feature differences at non-structurally connected part boundaries.

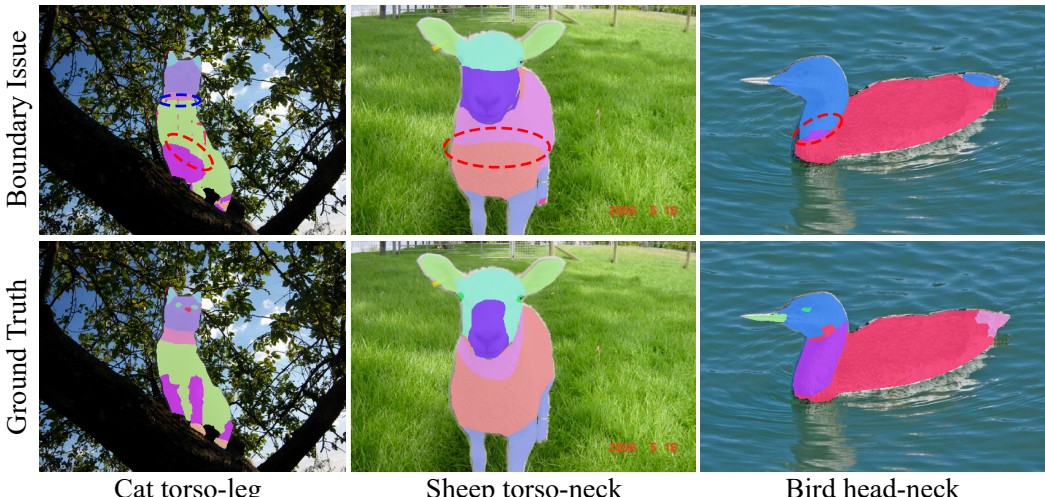

Figure 1: Boundary issues of existing methods on Pascal-Part-116 [21]. Dataset-specific part boundary definition variances limit the validity of pixel-alignment metrics for segmentation evaluation. For example, although the prediction boundary between cat neck and torso (PartCLIPSeg [28], blue dotted) deviates from annotations, it remains anatomically interpretable. Our method prioritizes anatomically/structurally implausible boundaries (red dotted).

The fundamental challenge in segmenting structurally connected part boundaries arises from the incompatibility between continuous, smooth image features and discrete classification mechanisms. Discrete classification enforces mutually exclusive label assignments and fails to account for pixels at structurally connected part boundaries, which exhibit hybrid characteristics of adjacent parts and lack clearly distinguishable features. To achieve accurate segmentation, it is essential to enhance the discrimination of these ambiguous features. Furthermore, since label assignment relies on category feature representations, adaptive optimization of category features can synergistically improve classification precision.

We propose the Boundary-Aware Refinement (BAR) module to refine the structurally connected part boundary. The BAR (1) first locates ambiguous boundary regions by analyzing the matching cost. In this work, we use the cosine similarity between the pixel features and part prototypes as the matching cost [29, 20]. The ambiguous regions are then separated from the original part masks, leaving the remaining deterministic regions as reliable references. (2) BAR then optimizes the pixel features within ambiguous regions by leveraging the context from deterministic regions and adaptively updates the part prototypes to better align with the current image characteristics. (3) Finally, ambiguous boundary regions are reclassified using enhanced pixel features and adapted prototypes.

In this paper, we propose Progressive Boundary-Aware Part Segmentation (PBAPS), a training-free OVPS framework that refines structurally connected part boundaries. Specifically, we first generate visual prototypes for each part class using Stable Diffusion [30], SAM [31], and DINOv2 [32]. Then, based on part structural relationships, a Hierarchical Part Connected Graph (HPCGraph) is constructed. Guided by this HPCGraph, progressive part segmentation is performed, where the BAR module mitigates boundary ambiguities and enhances segmentation precision.

In summary, the contribution of this paper is threefold: (i) We reveal the intrinsic cause of structurally connected part boundary errors: the conflict between continuous image features and discrete classification. (ii) We propose the BAR module, which improves boundary precision through feature optimization and dynamic prototype adaptation. (iii) We propose PBAPS, a novel and effective training-free OVPS method that integrates hierarchical reasoning with iterative boundary refinement, achieving state-of-the-art on Pascal-Part-116 [21], ADE20K-Part-234 [21], and PartImageNet [33].

## 2 Related Work

**Open Vocabulary Semantic Segmentation.** OVSS aims to overcome the limitations of predefined classes in traditional segmentation [34, 35, 36, 37, 38] by enabling zero-shot segmentation of unseen classes, requiring integration of the semantic understanding of VLM (e.g., CLIP [13], BLIP [14]) with pixel-level localization. Existing methods can be broadly categorized into two paradigms. One [10, 11, 16, 12, 17] is based on feature clustering and mask classification. OVSegmentor [39] leverages slot-attention [40] for pixel grouping and text alignment. RIM [41] employs image-to-image matching for training-free segmentation, constructing visual references and enhancing robustness via a relation-aware ranking distribution strategy. Our method generates part prototypes based on the process of constructing visual references in RIM. EBSeg [42] balances embeddings between base/novel classes and supplements spatial cues with SAM [31]. The second paradigm focuses on the alignment of the pixel-level features [18, 19, 20], which directly establishes associations between the vision and text features on the pixel scale. ZegCLIP [19] extends CLIP image-level classification capability to the pixel level. CAT-Seg [20] proposes a cost aggregation framework that optimizes CLIP image-text similarity through spatial and category aggregation.

**Open Vocabulary Part Segmentation.** Compared to OVSS, OVPS imposes higher demands on both model generalization and fine-grained recognition. Existing OVSS methods [43, 20, 21] often suffer significant performance degradation at the part level, mainly due to insufficient object-part structural modeling and limited fine-grained feature extraction. Recent studies have made notable progress. OPS [24] introduces class-agnostic part segmentation via object-aware spatial constraints and self-supervised feature optimization. ViRReq [25] decomposes part segmentation into composable atomic requests, leveraging a knowledge base for multigranular parsing. ViRReq and TAPPS [44] have previously explored object-to-part segmentation. In contrast, hierarchical reasoning in our method PBAPS introduces a cross-hierarchy matching mechanism, fully exploiting hierarchical context to strengthen feature discrimination. VLPart [45] constructs a joint vision-language embedding space through co-training, using DINO [46] dense semantic correspondence to parse novel objects into known components. In terms of benchmark construction, OV-PARTS [21] establishes the first OVPS benchmark (Pascal-Part-116, ADE20K-Part-234) with defined task scenarios. HIPIE [47] decouples hierarchical representations to separate foreground-background features. WPS-SAM [48] proposes a weakly supervised framework based on SAM [31], which reduces annotation dependency through co-training of learnable part prompt tokens and bounding box/point supervision. In feature enhancement, OIParts [27] fuses DINOv2 [32] local features with Stable Diffusion [30] global representations via adaptive channel selection. PartCLIPSeg [28] addresses fine-grained generalization by jointly optimizing separation and enhancement losses for part context modeling. PartCATSeg [49] improves semantic discrimination by constructing part-aware text embeddings combined with contrastive training, but lacks the ability to model structural relationships between parts.

## 3 Boundary Feature Gradient Analysis

To quantitatively analyze the feature variations across object boundaries, structurally connected part boundaries, and non-structurally connected part boundaries, we compute the spatial feature gradient. This feature gradient refers to the rate of spatial change in image feature vectors. Specifically, we extract pixel-wise features using DINOv2 [32]. For the feature vector $f(i,j) \in \mathbb{R}^{1 \times d}$ of each pixel $(i,j)$ in the image, we calculate the gradients in the horizontal $(x)$ and vertical $(y)$ directions for each channel $c$:

$$G_x(i,j,c) = |f(i+1,j,c) - f(i,j,c)|, \quad G_y(i,j,c) = |f(i,j+1,c) - f(i,j,c)| \qquad (1)$$

For each pixel $(i,j)$, we aggregate the gradients across all channels using the Euclidean norm to obtain the overall spatial feature gradient: $\text{Grad}(i,j) = \sqrt{\sum_{c=1}^{d} (G_x(i,j,c)^2 + G_y(i,j,c)^2)}$. A higher value of $\text{Grad}(i,j)$ corresponds to more pronounced local feature variation, whereas a lower value indicates smoother feature transitions. We extract these gradient values for all pixels located within the ground-truth masks of each boundary type and perform a systematic statistical analysis.

As shown in Table 1, the mean gradients and standard deviations (SD) exhibit consistent patterns across all three datasets: the object boundary shows the highest mean and SD values, while the

Table 1: Quantitative analysis of boundary feature gradients on Pascal-Part-116 [21], ADE20K-Part-234 [21], and PartImageNet [33]. The mean reflects the overall intensity of feature changes at the boundary. The standard deviation indicates the spatial consistency of boundary feature changes.

| Boundary Type | Pascal-Part-116 | | ADE20K-Part-234 | | PartImageNet | |
|---|---|---|---|---|---|---|
| | Mean | SD | Mean | SD | Mean | SD |
| Object | 0.5291 | 0.1300 | 0.5102 | 0.1253 | 0.5487 | 0.1352 |
| Non-structurally connected part | 0.5043 | 0.1164 | 0.4875 | 0.1108 | 0.5036 | 0.1204 |
| Structurally connected part | 0.4356 | 0.0762 | 0.4219 | 0.0721 | 0.4015 | 0.0658 |

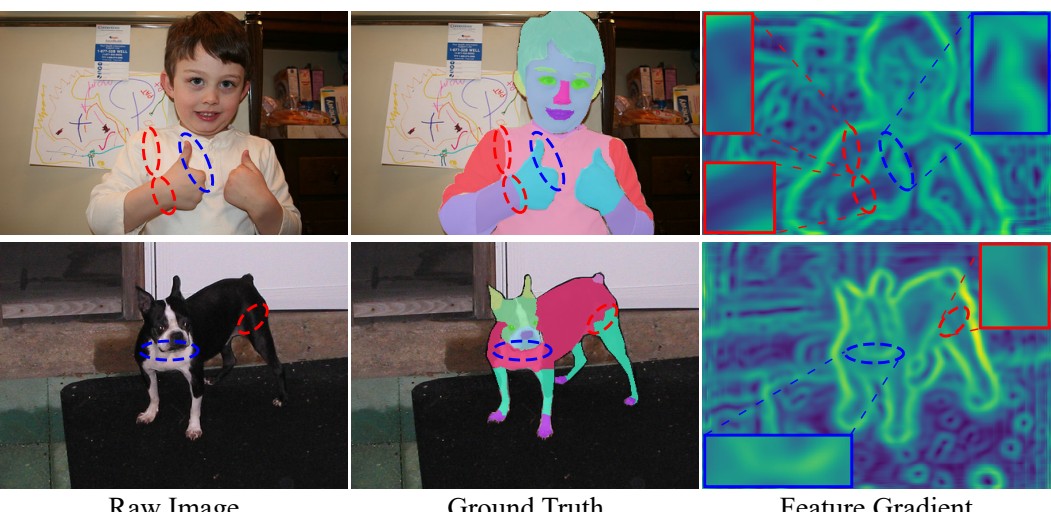

| Raw Image | Ground Truth | Feature Gradient |

Figure 2: Heatmap of feature gradients. (1) Object boundary: feature gradients at the boundaries between human/dog and other objects are the highest. (2) Non-structurally connected part boundary (blue dotted): these have notably higher gradients than interior regions, indicating clear local variations. (3) Structurally connected part boundary (red dotted): feature gradients are relatively low, reflecting smooth feature transitions between adjacent parts.

structurally connected part boundary presents the lowest values. This indicates that, regardless of the scale and diversity of the dataset, the object boundary exhibits the most significant feature differences and the most drastic changes. The structurally connected part boundary exhibits smooth feature transitions, while the non-structurally connected part boundary shows moderate feature changes, reflecting local differences. These results are consistent with Figure 2, provide theoretical support for optimizing structurally connected part boundaries in the BAR module.

The low feature gradients at structurally connected part boundaries reflect the conflict between the continuous feature space and the discrete semantic space. When pixel features exhibit smooth transitions between structurally connected parts, their feature vectors blend characteristics from both adjacent parts, making it difficult for discrete classification methods based on thresholds or similarity metrics to assign accurate labels. Moreover, the low feature gradients at these boundaries violate the common assumption that "boundaries have high gradients", rendering conventional boundary localization ineffective for such boundary.

To address these challenges, we design the Boundary-Aware Refinement (BAR) module, which explicitly locates ambiguous and deterministic regions at structurally connected part boundaries through cost-divergence maps. By leveraging context from deterministic regions, BAR enhances the discriminative features of ambiguous regions and adaptively optimizes part prototypes, thereby improving the segmentation accuracy of structurally connected part boundaries.

# 4    Method

Given an input image, the OVPS method is required to assign the correct part class label to each pixel. Unlike closed-set semantic segmentation, OVPS allows the test set $D_{test}$ to contain unseen part classes $C_{unseen}$ that are not included in the training set. In our training-free framework, all classes in the test set belong to $C_{unseen}$.

As illustrated in Figure 3, we present PBAPS, a training-free OVPS framework that operates in three stages: (1) generating visual prototypes via foundation models, (2) constructing HPCGraph based on structural prior knowledge, and (3) performing hierarchical segmentation guided by HPCGraph, during which the BAR module refines the structurally connected part boundaries.

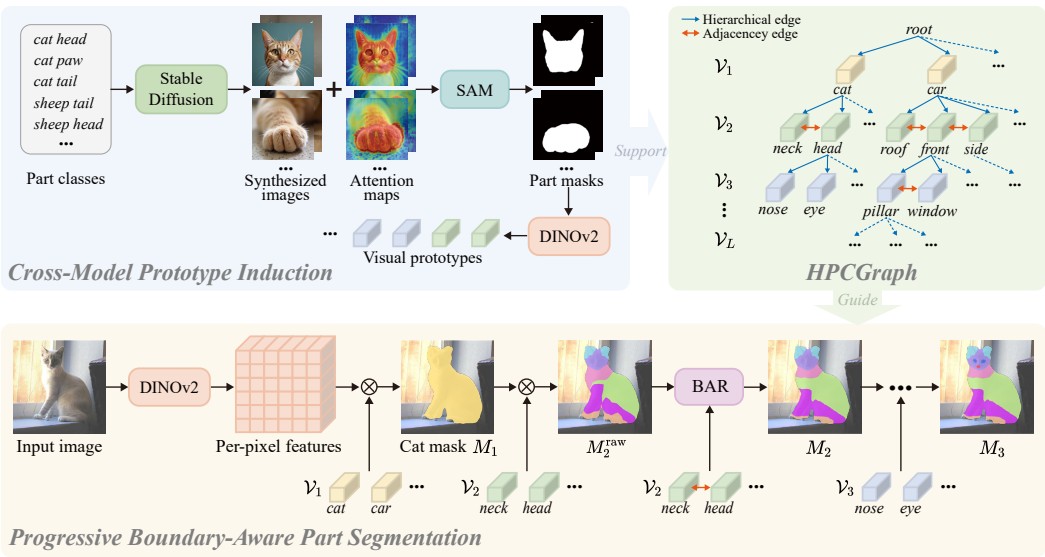

Figure 3: The architecture of PBAPS. Object-level segmentation is first performed through feature-prototype matching in $\mathcal{V}_1$, generating the object mask $M_1$. Taking the cat mask $M_1$ as an example, PBAPS then calculates part-level matching costs between the pixels and the corresponding cat part prototypes in $\mathcal{V}_2$, yielding the initial part mask $M_2^{\text{raw}}$. The BAR module subsequently refines structurally connected part boundaries in $M_2^{\text{raw}}$ to obtain the optimized part mask $M_2$.

## 4.1    Cross-Model Prototype Induction

It is important to note that the paradigm of semantic segmentation via visual prototype construction has been adopted and validated as effective in existing studies, such as OVDiff [50] and RIM [41]. The novelty of this work does not lie in the prototype construction itself, but rather in addressing the low segmentation accuracy of the structurally connected part boundaries in OVPS. Building upon the prototype-based paradigm, our method performs hierarchical segmentation guided by the HPCGraph, while the Boundary-Aware Refinement (BAR) module further improves boundary precision through feature optimization and dynamic prototype adaptation.

To generate visual prototypes for part classes, we follow the proven pipeline [41] that integrates Stable Diffusion [30], SAM [31] and DINOv2 [32]. (1) First, Stable Diffusion generates synthetic images for each part $c$ in $D_{test}$ using text prompts (e.g., "a photo of $c$"), while extracting cross-attention maps to localize the target part $c$ across multiple layers and timesteps [51, 50, 41]. These attention maps are processed via normalization and thresholding to generate a high-confidence region mask $M$. (2) Within regions where $M = 1$, we randomly sample $k$ foreground prompt points for SAM, which produces a binary mask corresponding to part $c$. (3) Finally, DINOv2 extracts features from the masked regions of the synthetic images, and the visual prototype $\mathbf{p}_c \in \mathbb{R}^{1 \times d}$ for $c$ is computed as the average feature vector in these regions, serving as a global semantic representation for subsequent matching and boundary refinement.

## 4.2 Hierarchical Part Connected Graph

We construct the Hierarchical Part Connected Graph (HPCGraph) based on part structural relationships. The HPCGraph is defined as $G = (\mathcal{V}, \mathcal{E})$, where the node set $\mathcal{V} = \{\mathcal{V}_0, \mathcal{V}_1, \ldots, \mathcal{V}_L\}$ consists of $L + 1$ hierarchical layers. Specifically, $\mathcal{V}_0 = \{\text{root}\}$ serves as the starting symbol, $\mathcal{V}_1$ denotes the object nodes, and $\mathcal{V}_l$ ($l > 1$) represents the part nodes of different granularity. The edge set $\mathcal{E}$ comprises two types of relationships: (1) Hierarchical edge $e_{p \to c}$ from the parent node $v_p \in \mathcal{V}_l$ to the child node $v_c \in \mathcal{V}_{l+1}$, representing the "composition" relationship (e.g., "cat head" → "cat eye"). (2) Adjacency edge $e_{i-j}$ between nodes $v_i, v_j \in \mathcal{V}_l$ if they have structural connection (e.g., "cat head" ↔ "cat neck").

**Cross-hierarchy Matching Cost.**    To effectively exploit object-part and part-part relationships, we define the matching cost between a pixel feature $f \in \mathbb{R}^{1 \times d}$ and a node $v \in \mathcal{V}$ as the maximum cosine similarity within its dominance set $\mathcal{D}(v) = \{v\} \cup \{\textit{descendant nodes of } v\}$:

$$S(f, v) = \max_{u \in \mathcal{D}(v)} \left[ \frac{f^\top \mathbf{p}_u}{|f| \cdot |\mathbf{p}_u|} \right] \tag{2}$$

## 4.3 Progressive Boundary-Aware Part Segmentation

Given an input image $I \in \mathbb{R}^{h \times w \times 3}$, we extract pixel features $F \in \mathbb{R}^{h \times w \times d}$ using DINOv2, adopting a sliding-window strategy [52] to preserve fine spatial details. The progressive part segmentation is defined as $Segment(F, \mathcal{V}_l) \to M_l$, where $\mathcal{V}_l$ denotes the node set of the layer $l$ and $M_l$ represents the corresponding segmentation mask produced for the layer $l$.

**Object-level Segmentation ($l = 1$).**    For each pixel feature $f(i, j)$ in the feature map $F$, we calculate cross-hierarchy matching costs with all nodes in $\mathcal{V}_1$. The resulting object-level segmentation mask $M_1$ is obtained by assigning to each pixel $(i, j)$ the node with the highest matching score: $M_1(i, j) = \arg\max_{v \in \mathcal{V}_1} S\left(f(i, j), v\right)$.

**Part-level Segmentation ($l \geq 2$).**    For each parent node $p \in \mathcal{V}_{l-1}$, perform localized segmentation within its corresponding mask $M_{l-1}(p)$. Specifically, we extract the parent-region feature $F_p = F \odot M_{l-1}(p)$ and calculate the cross-hierarchy matching costs between $F_p$ and its child nodes $C \subseteq \mathcal{V}_l$ to generate raw mask $M_{l,p}^{\text{raw}} = \arg\max_{c \in C} S(F_p, c)$. The overall segmentation mask at layer $l$ is $M_l^{\text{raw}} = \bigcup_{p \in \mathcal{V}_{l-1}} M_{l,p}^{\text{raw}}$. Subsequently, the BAR module optimizes structurally connected part boundaries through feature optimization and prototype adaptation, generating the final refined masks $M_l$. The progressive part segmentation ends when the current layer $l$ contains only atomic parts (i.e., $\forall v \in \mathcal{V}_l, C(v) = \emptyset$).

## 4.4 Boundary-Aware Refinement

The BAR module refines structurally connected part boundaries via four steps, as shown in Figure 4.

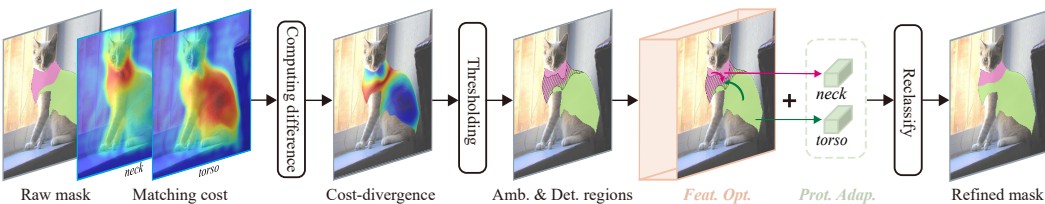

Figure 4: Example of Boundary-Aware Refinement. When segmenting the cat neck and torso, the corresponding matching-cost maps are obtained. Taking their absolute difference quantifies the classification ambiguity, where salient regions indicate similar matching costs. Thresholding this difference map yields ambiguous regions (shaded areas), and the rest as deterministic regions. Ambiguous-region feature optimization and prototype adaptation are then applied to refine the boundary between the cat neck and torso.

**Ambiguous Boundary Region Localization.** To address the uncertainty at structurally connected part boundaries, we first quantify the classification ambiguity using a cost-divergence map defined as $D_{A,B} = |S_A - S_B|$, where $S_A, S_B \in \mathbb{R}^{h \times w}$ denote the matching cost maps of the structurally connected parts $A$ and $B$. The low divergence values correspond to regions where both parts exhibit similar matching costs, indicating semantic ambiguity. We normalize $D_{A,B}$ and apply a threshold to identify the ambiguous region $U = \mathbb{I}(\text{Norm}(D_{A,B}) \leq \lambda)$, where $\lambda \in [0, 1]$ is the ambiguity threshold. The high-confidence deterministic regions are then obtained by excluding $U$ from the initial part masks $A_{\text{raw}}, B_{\text{raw}}$:

$$A_{\text{det}} = A_{\text{raw}} \odot (1 - U), \quad B_{\text{det}} = B_{\text{raw}} \odot (1 - U) \tag{3}$$

**Ambiguous Pixel Feature Optimization.** The deterministic regions serve as reliable context for refining ambiguous pixels, as their features are distinct and free from classification uncertainty. For each ambiguous pixel $m \in U$, we enhance its discriminative characteristics by incorporating the context from deterministic regions $A_{\text{det}}$ and $B_{\text{det}}$. Specifically, we first compute attention weights $w_A(m, a)$ by normalizing cosine similarities between the ambiguous pixel feature $f_m$ and all pixels $a \in A_{\text{det}}$. These weights quantify the relevance of each pixel in $A_{\text{det}}$ to the ambiguous pixel $m$. Using $w_A(m, a)$ to aggregate contextual features from $A_{\text{det}}$:

$$w_A(m, a) = \frac{\exp(\cos(f_m, f_a))}{\sum_{e \in A_{\text{det}}} \exp(\cos(f_m, f_e))}, \quad \mathbf{c}_A(m) = \sum_{a \in A_{\text{det}}} w_A(m, a) f_a \tag{4}$$

Similarly, we compute $w_B(m, b)$ and aggregate the context of $B_{\text{det}}$ into $\mathbf{c}_B(m)$. The optimized feature $\tilde{f}_m$ is then derived by fusing the original feature $f_m$ with the contextual feature of both parts, this fusion mitigates the ambiguity of hybrid features by "pulling" $f_m$ toward the discriminative characteristics of deterministic regions, thereby improving feature separability and boundary precision:

$$\tilde{f}_m = \gamma \cdot f_m + (1 - \gamma) \cdot \frac{\mathbf{c}_A(p) + \mathbf{c}_B(p)}{2} \tag{5}$$

**Visual Prototype Adaptive Refinement.** To adapt global part knowledge to the specific image context, we refine visual prototypes by integrating global priors with local image features. For the deterministic region of part $A$, we cluster its features $F_{\text{det}}^A = \{f_a \mid a \in A_{\text{det}}\}$ using K-means to obtain a local prototype $\mathbf{q}_A$, which captures the dominant appearance patterns of part $A$ in the current image. The adaptive prototype is then computed by fusing the global prototype $\mathbf{p}_A$ with its corresponding local prototype $\mathbf{q}_A$:

$$\tilde{\mathbf{p}}_A = \alpha \cdot \mathbf{p}_A + (1 - \alpha) \cdot \mathbf{q}_A \tag{6}$$

where $\alpha \in [0, 1]$. This adaptation allows $\tilde{\mathbf{p}}_A$ to preserve the universal characteristics of part $A$ while incorporating image-specific variations (e.g., pose, texture), making it more relevant for matching ambiguous pixels in the current scene. Similarly, obtain $\tilde{\mathbf{p}}_B$ for part $B$.

Notably, the similarity between synthetic prototypes and real test images can affect segmentation performance. Our approach achieves a balance between effectiveness and generalization by employing synthetic prototypes with moderate similarity and combining them with the dynamic prototype adaptation mechanism. This design maintains high segmentation accuracy without overfitting to specific datasets, enabling robust performance across diverse domains. Moreover, even when the attention-based masks derived from Stable Diffusion contain minor inaccuracies (e.g., small non-target regions), the prototype adaptation mechanism effectively mitigates such issues. By combining global prototypes with contextual features from deterministic regions, the resulting prototypes dynamically align with the true object parts present in the image, thus enhancing segmentation precision and robustness.

**Ambiguous Region Reclassification.** For each ambiguous pixel $m \in U$, we perform a reclassification using the optimized pixel feature $\tilde{f}_m$ and the adaptive prototypes $\tilde{\mathbf{p}}_A, \tilde{\mathbf{p}}_B$:

$$y(m) = \arg \max_{c \in \{A, B\}} \cos(\tilde{f}_m, \tilde{\mathbf{p}}_c) \tag{7}$$

# 5 Experiments

## 5.1 Datasets and Evaluation

We evaluate PBAPS on three benchmarks: (1) Pascal-Part-116 [21], which refines the Pascal-Part [53] by merging over-segmented parts and removing redundant descriptors. The validation set includes 17 object classes, 116 part classes, and 850 images. (2) ADE20K-Part-234 [21], derived from ADE20K [54] via low-frequency class filtering and synonym merging, containing 44 object classes, 234 part classes, and 1016 validation images. (3) PartImageNet [33], which groups 158 ImageNet [55] classes into 11 superclasses with uniform part structures, follows prior work [28] to evaluate 40 common object categories on 2957 validation images. In line with previous works [45, 28], we use the mean Intersection over Union (mIoU) to assess overall segmentation quality. Additionally, the boundary Intersection over Union (bIoU) [56] is introduced to specifically evaluate the accuracy of part boundary.

## 5.2 Implementation Details

**Visual Prototype Generation.** For each part class, we generate hundreds of 512×512 synthetic images using Stable Diffusion v1.4 [30], along with their corresponding cross-modal attention maps. After binarizing attention maps (threshold=0.7), 5 prompt points are randomly sampled for ViT-B SAM [31] to obtain the corresponding part masks. The features within the masked regions are then extracted using ViT-B DINOv2 [32]. Finally, K-means clustering ($K = 4$) is applied to these features to construct subcategory prototypes, thus capturing intra-class morphological diversity and improving the robustness of the part prototype.

**Model Inference.** We extract pixel-wise features from input images using ViT-B DINOv2 [32] with a sliding-window strategy [52], using the "key" values from the final attention layer as feature representations [57]. The hyperparameters of the BAR module are fixed as follows: ambiguity threshold $\lambda_{amb} = 0.3$, feature fusion weight $\gamma = 0.8$, and prototype adaptation coefficient $\alpha = 0.7$.

## 5.3 Comparison with Existing Methods

Table 2: Comparison with existing methods. * denotes our re-implementation. Bold and underline indicate the best and second-best results, respectively.

| Method | Backbone | Supervision | Zero-shot transfer | Pascal-Part-116 | | ADE20K-Part-234 | | PartImageNet | |
|---|---|---|---|---|---|---|---|---|---|
| | | | | mIoU | bIoU | mIoU | bIoU | mIoU | bIoU |
| ZSSeg+ [16, 21] | ResNet-50 | class label | ✗ | 24.91 | 18.18 | 19.84 | 12.89 | - | - |
| VLPart [45] | ResNet-50 | class label | ✗ | 25.98 | 16.79 | - | - | - | - |
| CLIPSeg [43] | ViT-B/16 | class label | ✗ | 24.23 | 15.98 | 5.88 | 4.87 | 26.98 | 18.54 |
| CAT-Seg [20] | ViT-B/16 | class label | ✗ | 30.53 | 21.25 | 8.88 | 7.71 | 28.56 | 20.21 |
| PartCATSeg [49] | ViT-B/16 | class label | ✗ | 29.54 | 20.96 | 14.68 | 11.99 | 30.18 | 21.29 |
| PartCLIPSeg [28] | ViT-B/16 | class label | ✗ | 35.96 | 26.72 | 12.64 | 9.69 | 30.38 | 21.04 |
| OVDiff [50] | UNet | Training-free | | 41.55 | 31.57 | 22.13 | 15.83 | 38.22 | 26.71 |
| RIM* [41] | UNet+ViT-B/16 | Training-free | | 43.19 | 32.13 | 20.91 | 13.30 | 39.32 | 25.06 |
| PBAPS (ours) | UNet+ViT-B/16 | Training-free | | **46.35** | **34.46** | **24.70** | **16.41** | **42.61** | **29.31** |

We compare PBAPS with ZSSeg+, VLPart, CLIPSeg, CAT-Seg, PartCATSeg and PartCLIPSeg, which are fine-tuned on target benchmarks. As shown in Table 2, PBAPS outperforms the state-of-the-art full-supervised method PartCLIPSeg [28] by more 10% in mIoU across the three datasets, with bIoU improvements of 7.74%, 6.72% and 8.27% on Pascal-Part-116, ADE20K-Part-234, and PartImageNet, respectively. We also compare PBAPS with training-free OVSS methods OVDiff [50] and RIM [41], using identical visual prototype set for fairness. All three visual prototype matching methods outperform finetuning-based methods, validating the effectiveness of this paradigm in OVPS. Our method achieves state-of-the-art mIoU and bIoU scores on all three datasets, surpassing OVDiff and RIM. Figure 5 visually demonstrates its superiority. Unlike RIM, which under-segments when target parts lack distinct boundaries, PBAPS captures subtle feature transitions between connected parts, enhancing segmentation accuracy.

## 5.4 Analysis and Ablation Study

Table 3: Ablation study for PBAPS on Pascal-Part-116. The baseline model (1st row) excludes all components, directly matching pixel-wise features extracted by DINOv2 [32] with visual prototypes.

| HPCGraph | | BAR | | | Pascal-Part-116 | |
|---|---|---|---|---|---|---|
| w/ Hier. Seg. | w/ X-Hier. Match | w/ Boun. Loc. | w/ Feat. Opt. | w/ Proto. Adapt. | mIoU | bIoU |
| | | | | | 40.74 | 31.09 |
| ✓ | | | | | 43.37 | 32.66 |
| ✓ | ✓ | | | | 44.08 | 32.99 |
| ✓ | ✓ | | ✓ | | 43.86 | 32.89 |
| ✓ | ✓ | | | ✓ | 44.83 | 33.42 |
| ✓ | ✓ | ✓ | ✓ | | 45.58 | 34.04 |
| ✓ | ✓ | ✓ | | ✓ | 45.79 | 34.16 |
| ✓ | ✓ | ✓ | ✓ | ✓ | **46.35** | **34.46** |

**Ablation study for HPCGraph.** The HPCGraph enhances part-level semantic modeling via hierarchical segmentation and cross-hierarchy matching. As shown in Table 3, incorporating hierarchical segmentation (2nd row) increases mIoU and bIoU by 2.63% and 1.57%, respectively, indicating that modeling top-down part relationships effectively improves segmentation consistency. Further integration of cross-hierarchy matching (3rd row) raises mIoU and bIoU to 44.08% and 32.99%, demonstrating that cross-level prototype comparison strengthens fine-grained part discrimination and improves boundary precision.

**Ablation study for BAR.** As shown in Table 3, using feature optimization alone without ambiguous boundary localization (4th row) leads to slight performance degradation, mIoU and bIoU drop by 0.22% and 0.1% compared to the 3rd row, indicating that indiscriminate context fusion weakens local discriminative features. Introducing boundary region localization (6th row) improves mIoU and bIoU by 1.5% and 1.05%, validating the importance of distinguishing deterministic from ambiguous regions. Moreover, the 5th and 7th rows demonstrate the synergistic effect between prototype adaptation and boundary localization, which jointly improve boundary precision. When all components are integrated (8th row), mIoU and bIoU increase by 5.61% and 3.37% over the baseline, confirming the overall effectiveness of the BAR module in refining part boundaries.

**Effectiveness of boundary region localization.** As shown in Table 4, we assess the boundary localization capability of the BAR module by computing boundary recall, which measures the proportion of true boundary pixels correctly covered by the detected ambiguous regions. When the ambiguity threshold is set to $\lambda_{amb} = 0.3$, the model achieves the optimal balance between the boundary recall (78.93%) and the segmentation precision (46.35% mIoU). A strict threshold ($\lambda_{amb} = 0.1$) filters out 64.78% of true boundary pixels, leading to insufficient refinement, while a lenient threshold ($\lambda_{amb} = 0.5$) improves recall to 88.97% but introduces non-boundary noise, reducing mIoU and bIoU.

Table 4: Pascal-Part-116 results with different ambiguity threshold.

| $\lambda_{amb}$ | Boundary Recall | mIoU | bIoU |
|---|---|---|---|
| 0.1 | 35.22 | 44.48 | 33.42 |
| 0.2 | 58.70 | 45.54 | 33.81 |
| 0.3 | 78.93 | **46.35** | **34.46** |
| 0.4 | 82.15 | 45.71 | 34.01 |
| 0.5 | **88.97** | 44.29 | 33.10 |

Table 5: Impact of part boundary optimization types on Pascal-Part-116.

| Optimization Type | mIoU | bIoU | Inference Time (s) |
|---|---|---|---|
| Baseline | 44.08 | 32.99 | **1.47** |
| Non-struct. boundary | 43.19 | 32.48 | 2.06 |
| All part boundary | 45.43 | 33.95 | 2.19 |
| Struct. boundary | **46.35** | **34.46** | 1.81 |

**Impact of BAR for different boundary.** As shown in Table 5, we analyze the impact of the BAR module on different types of part boundaries. The baseline (1st row) employs the HPCGraph without BAR. For structurally connected part boundaries, BAR achieves the most significant improvements (2.27% mIoU, 1.47% bIoU). In contrast, applying BAR to non-structurally connected part boundaries (2nd row) slightly decreases both mIoU and bIoU, while increasing inference time by 0.25s compared

to the 4th row. This degradation occurs because such boundaries are already well separated, and additional context fusion introduces noise that blurs the boundaries. Moreover, the greater number of boundaries increases the overall inference time. Although applying BAR to all part boundaries improves mIoU by 1.35%, it increases the inference time by 48%, confirming that selective optimization of structurally connected boundaries offers the best balance between accuracy and efficiency.

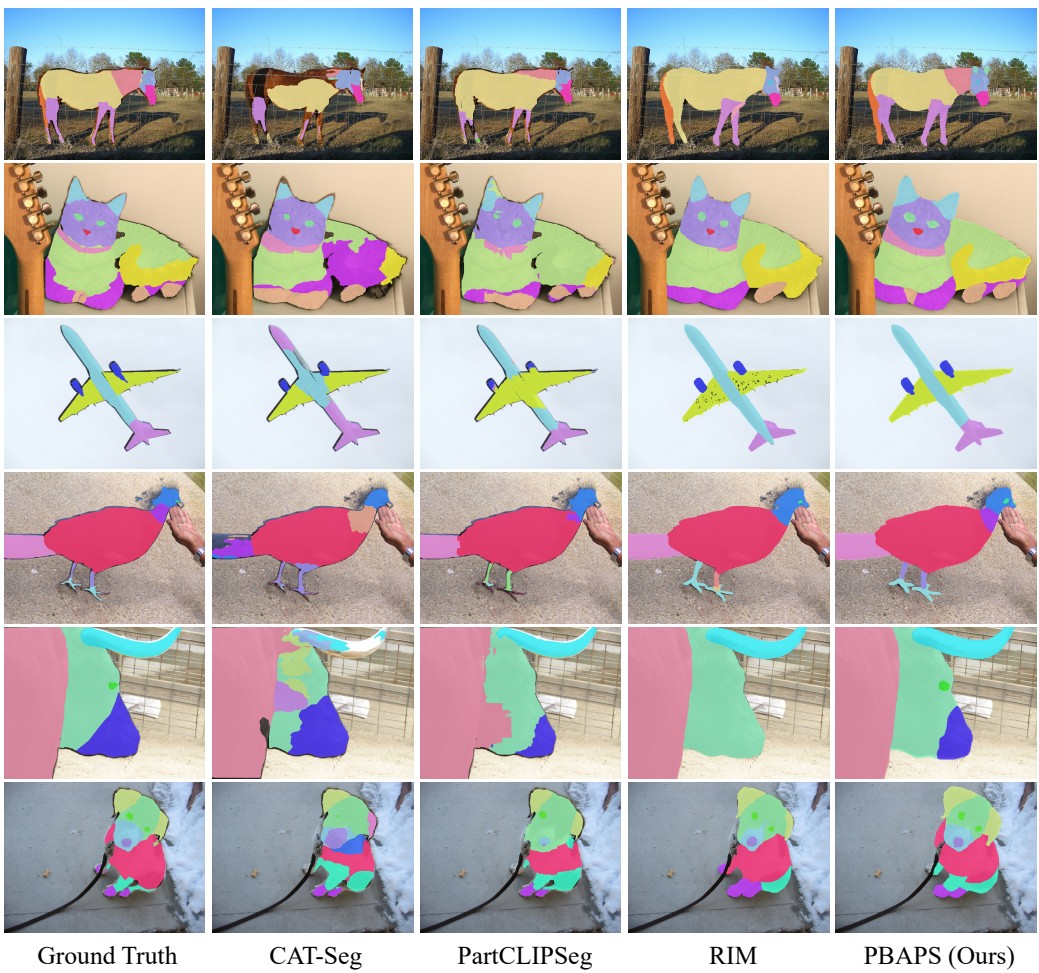

| Ground Truth | CAT-Seg | PartCLIPSeg | RIM | PBAPS (Ours) |

Figure 5: Qualitative results on Pascal-Part-116.

## 6 Conclusion

This study addresses the inaccurate segmentation of structurally connected part boundaries by introducing PBAPS, a training-free OVPS framework. PBAPS integrates a progressive segmentation strategy guided by HPCGraph with a BAR module to enhance boundary precision. Extensive experiments demonstrate that PBAPS consistently outperforms state-of-the-art methods across multiple benchmarks, confirming its effectiveness and generalizability. This study offers a novel and interpretable solution for fine-grained part segmentation in open-world scenarios.

## 7 Acknowledgement

This research was supported by the National Natural Science Foundation of China (No.62476192) and the Natural Science Foundation of Tianjin (No.23JCQNJC02010). The support of these foundations has been instrumental in advancing this study, for which we express our sincere gratitude.

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

# A Discussion

## A.1 Limitations and Future Work

Although our method shows excellent performance in the OVPS task, it still has several limitations. First, PBAPS focuses on part-level semantic segmentation and cannot distinguish different instances of the same category. For example, when multiple cats appear in an image, PBAPS segments all "cat head" without assigning each head to its respective individual. Second, our method relies on the HPCGraph, which defines hierarchical and adjacency relationships between objects and parts via manual design or external knowledge bases. However, when processing objects with complex, irregular structures (e.g., modular furniture), the statically predefined graph may fail to capture their intrinsic part topology and semantic associations.

To address these limitations, future work can explore two directions. On the one hand, introducing instance-aware mechanisms could enable simultaneous semantic segmentation and instance separation. On the other hand, developing a dynamic hierarchical reasoning framework that can adaptively adjust part hierarchical structures based on specific objects.

## A.2 Social Impact

Our method offers significant societal value in multiple domains. In healthcare, its precise segmentation of structurally connected parts can improve the efficiency of pathological diagnosis and provide a low-cost, annotation-free diagnostic aid for remote regions, thereby promoting equitable access to medical resources. In industrial manufacturing, its hierarchical part reasoning capability supports automated quality inspection and disassembly of mechanical equipment.

Despite these advantages, PBAPS also has potential risks. In safety-critical applications such as medical imaging and autonomous driving, reliance on predefined structural priors may lead to segmentation errors in complex scenarios. For example, misidentifying the boundaries of rare pathological organs, causing diagnostic inaccuracies.

# B Additional Experiments

## B.1 Further Details

**Stable Diffusion.** A text-conditioned latent diffusion model [30] that generates images via iterative denoising and has three main components: (1) a pre-trained VAE [58] for image encoding/decoding, (2) a text encoder $\tau$ converting the prompt into embedding vector, and (3) a time-conditional U-Net $\phi$ that denoises an initial Gaussian noise to produce the image. During image synthesis, the corresponding cross-attention maps $A$ [51] are extracted to locate the target part. Specifically, the prompt $p$ is encoded in text embedding $\tau(p) \in \mathbb{R}^{N \times d}$, where $N$ denotes the length of the token sequence. At each U-Net $\phi$ timestep $t$, visual features $\phi(z_t) \in \mathbb{R}^{H \times W \times C}$ from a noisy image $z_t$ are flattened and projected to query $Q$, while text embedding $\tau(p)$ produces the key $K$ and the value $V$ via learnable layers $\ell_K$ and $\ell_V$:

$$Q = \ell_Q(\phi(z_t)), \quad K = \ell_K(\tau(p)), \quad V = \ell_V(\tau(p)) \tag{8}$$

The cross-attention weights $A = \text{Softmax}\left(\frac{QK^T}{\sqrt{d}}\right) \in \mathbb{R}^{H \times W \times N}$. To obtain robust class attention maps, attention maps are aggregated over multiple layers and time steps:

$$\tilde{A}^j = \frac{1}{|B||T|} \sum_{b \in B} \sum_{t \in T} \frac{A_{b,t}^j}{\max(A_{b,t}^j)} \tag{9}$$

where $j$ is the index of the text token, $B$ and $T$ denote the layers and timestep sets, respectively.

**Segment Anything Model.** SAM [31] is a training-free image segmentation framework that enables prompt-based rapid segmentation of arbitrary objects using point, box, or mask input. Its architecture consists of: (1) an image encoder that extracts the global visual feature $F_i$ from the input image, (2) a prompt encoder that encodes prompts into unified features $F_p$, and (3) a mask decoder that generates candidate masks by integrating $F_i$ with $F_p$.

**DINOv2.** DINOv2 [32] is a discriminative self-supervised ViT that distills general visual features from large-scale unlabeled data. Through joint global-image and local-patch level learning, its robust features excel in tasks including image classification, semantic segmentation, and patch matching.

For each part class $c$, we mainly adopt the generic prompt template "a photo of $c$" to guide Stable Diffusion [30] in synthesizing images of the part $c$. To further enrich visual diversity, we incorporate synonyms and subclasses of the original class names [51]. We also standardize class names (e.g., "tvmonitor" $\rightarrow$ "tv monitor") and refine them to resolve ambiguities (e.g., "cat hand" $\rightarrow$ "cat paw").

During image generation, we set an independent random seed for each class to ensure reproducibility. The image generation time scales linearly with the number of images: 124s for 32 images, 227s for 64 images, and 443s for 128 images. Each part prototype, extracted via DINOv2 [32], occupies 0.45 MB. During inference, PBAPS employs a sliding window of size 224×224 with a stride of 64 for feature extraction using DINOv2 (identical to all baseline methods), achieving an average processing time of 1.81s per image. All results are measured with a single NVIDIA A6000 GPU.

### B.2 Ablation on Feature Extractors

Table 6: Comprison of different feature extractors on Pascal-Part-116 [21].

| Feature Extractor | mIoU | bIoU |
|---|---|---|
| MAE [59] | 37.94 | 27.47 |
| DINO [60] | 41.20 | 30.79 |
| CLIP [13] | 42.39 | 30.41 |
| Stable Diffusion [30] | 44.12 | 33.31 |
| DINOv2 [32] | **46.35** | **34.46** |

Table 7: Comprison of mask generators on Pascal-Part-116 [21].

| Mask Generator | mIoU | bIoU |
|---|---|---|
| MaskFormer [61] | 41.28 | 30.97 |
| SAM [31] | 43.82 | 32.43 |
| None | **46.35** | **34.46** |

Our method can integrate with any pretrained visual feature extractor to construct visual prototypes and extract image features. As shown in Table 6, to demonstrate the superiority of using DINOv2 [32] for image feature extraction in our framework, we compare it with several self-supervised ViT feature extractors. DINOv2 significantly outperforms other methods, benefiting from its pretraining based on image-level and patch-level discriminative learning, which empowers it with fine-grained feature representation capabilities. CLIP exhibits limitations in fine-grained feature alignment, likely due to its contrastive learning that focuses on global feature alignment (Figure 6). MAE yields the weakest performance due to its lack of explicit semantic discriminative learning. Stable Diffusion performs secondarily by leveraging structural information implicitly learned through generative tasks.

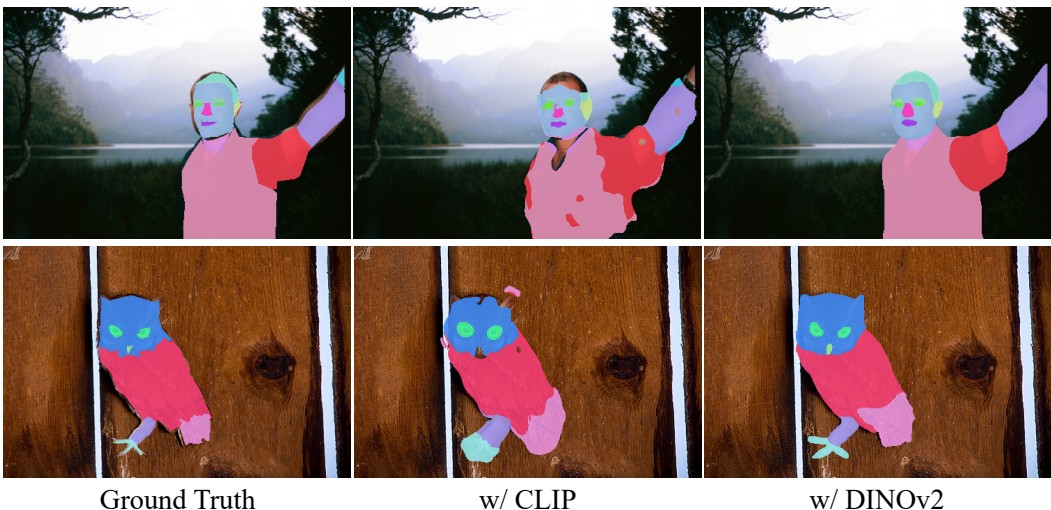

Ground Truth         w/ CLIP         w/ DINOv2

Figure 6: Qualitative ablation on feature extractors in Pascal-Part-116 [21].

## B.3 Ablation on Mask Generators

Our PBAPS framework supports two segmentation paradigms: (1) direct per-pixel feature classification and (2) region-level classification following mask generation by a pre-trained segmenter. To validate the superiority of our pixel-wise classification strategy, we compare it against MaskFormer-based, SAM-based variants. Note that when using MaskFormer [61] or SAM [31] for mask generation, the segmentation process operates only under the structural constraints of HPCGraph without boundary refinement via the BAR module, as these generators only produce binary masks, lacking class-specific matching cost maps. As shown in Table 7, introducing MaskFormer or SAM significantly degrades performance. This is because MaskFormer, trained on object-level COCO [62, 8, 63], tends to propose coarse object regions and fails to capture subtle part-level distinctions. Although SAM exhibits zero-shot generalization, its segmentation relies on prominent visual changes (e.g., texture, color). The part-level segmentation often involves only local, subtle variations, leading to frequent under-segmentation by SAM (Figure 7).

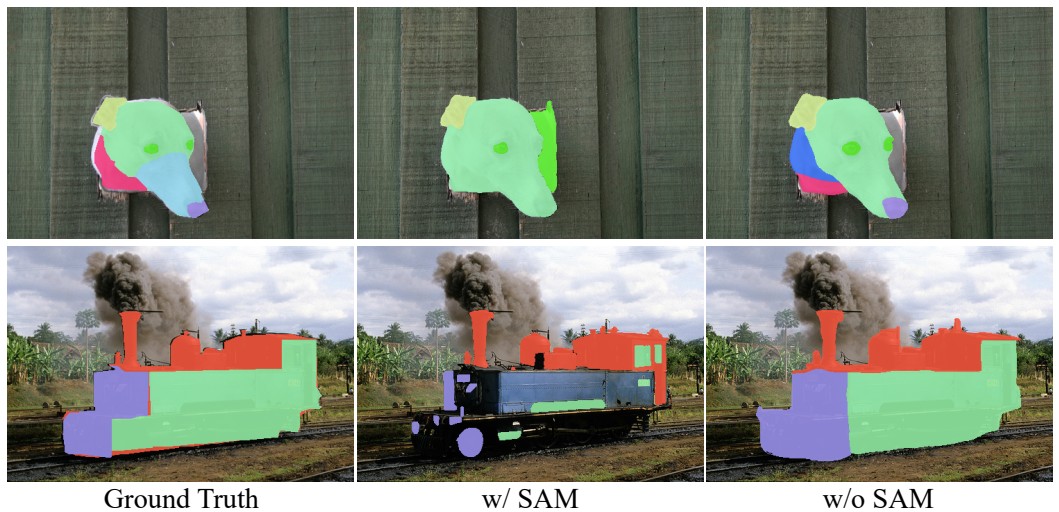

| Ground Truth | w/ SAM | w/o SAM |

Figure 7: Qualitative ablation on mask generators in Pascal-Part-116 [21].

## B.4 Sensitivity Analysis of Hyperparameters

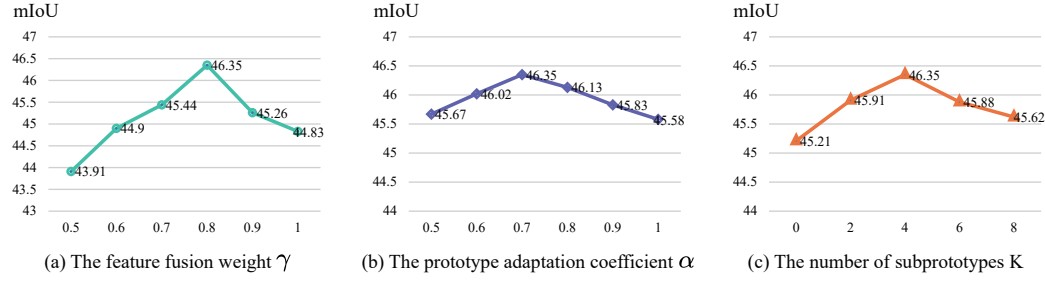

(a) The feature fusion weight $\gamma$     (b) The prototype adaptation coefficient $\alpha$     (c) The number of subprototypes K

Figure 8: The sensitivity analysis of $\gamma$, $\alpha$ and $K$ on Pascal-Part-116 [21].

To evaluate the impact of key hyperparameters in PBAPS, we conduct a sensitivity analysis on Pascal-Part-116 for three core parameters: feature fusion weight $\gamma$, prototype adaptation coefficient $\alpha$, and the number of subprototypes $K$ in visual prototype generation. Figure 8 shows that when $\gamma = 0.8$, PBAPS achieves the highest mIoU of 46.35%. A moderate $\gamma$ balances the retention of critical original feature information and the integration of discriminative context from deterministic regions. When $\gamma$ is low (0.5), excessive reliance on contextual features suppresses original discriminative information, degrading performance. When $\gamma$ is high (1.0), the neglect of contextual guidance leaves

feature ambiguity unresolved. Similarly, a moderate $\alpha$ (0.7) enables adaptive prototypes to retain both universal part knowledge and image-specific variations (e.g., pose, texture). When $\alpha$ is small, over-adaptation to local features may introduce noise or cause overfitting to image details. When $\alpha$ approaches 1.0, the prototypes lack adaptability to variations in the current image. Regarding the number of subprototypes, $K = 4$ allows the model to fully capture intra-class diversity while avoiding noise interference. When $K = 0$, the global prototype is overly generic; when $K > 4$, excessive subprototypes may introduce noisy clusters, leading to performance degradation.

## B.5 Additional Qualitative Results

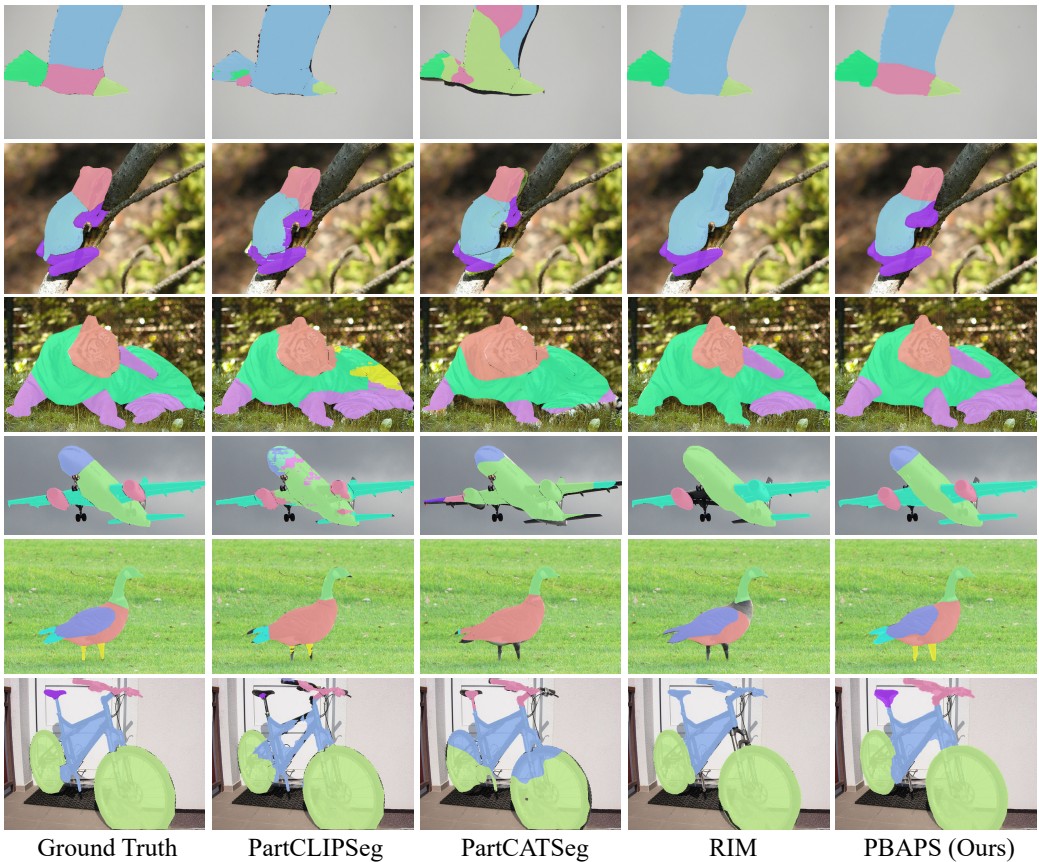

|  |  |  |  |  |
| --- | --- | --- | --- | --- |
| Ground Truth | PartCLIPSeg | PartCATSeg | RIM | PBAPS (Ours) |

Figure 9: Qualitative results on PartImageNet [33].

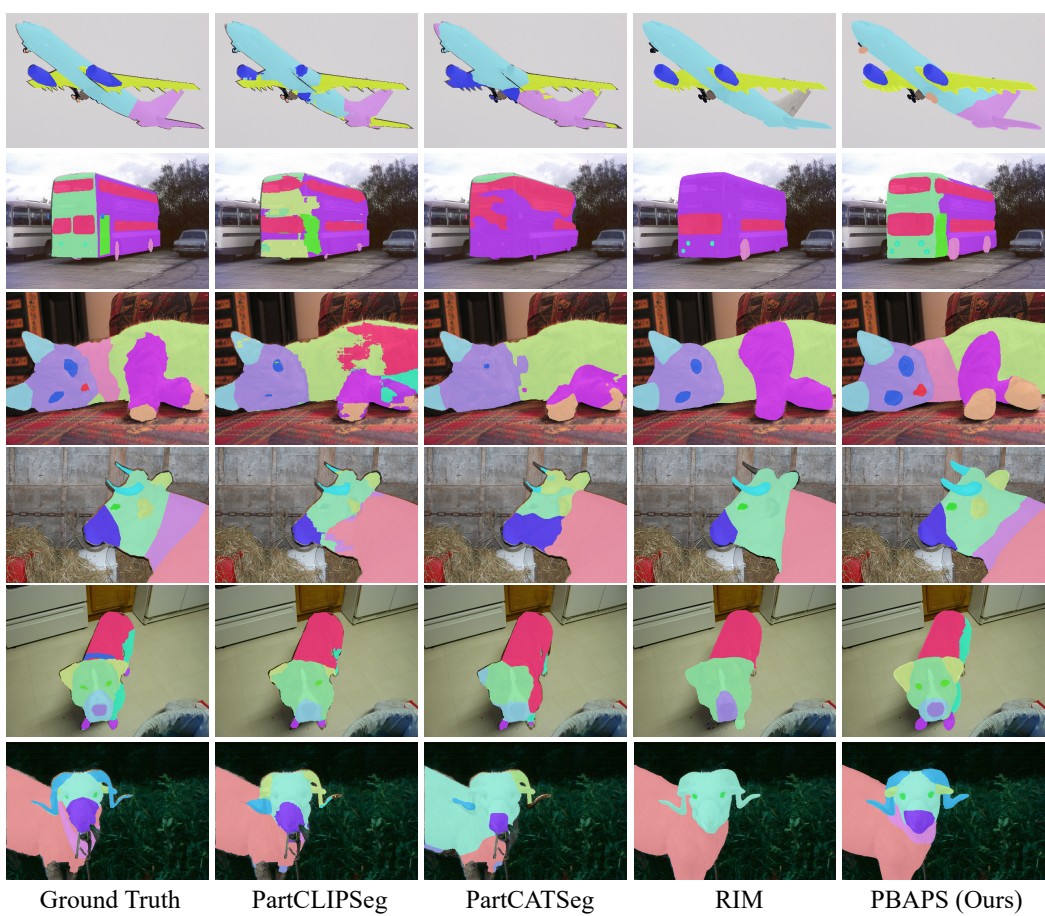

| Ground Truth | PartCLIPSeg | PartCATSeg | RIM | PBAPS (Ours) |
| --- | --- | --- | --- | --- |

Figure 10: Qualitative results on Pascal-Part-116 [21].

