# OpenReview forum: "Open-Vocabulary Part Segmentation via Progressive and Boundary-Aware Strategy"
_NeurIPS.cc/2025/Conference — NeurIPS 2025 poster_

### Official Review · Reviewer_4fTr · 2025-06-26

**Clarity:** 2
**Significance:** 2
**Originality:** 2
**Rating:** 4
**Confidence:** 4

**Summary:**

This paper presents Progressive Boundary-Aware Part Segmentation (PBAPS), a training-free model for open-vocabulary part segmentation (OVPS). This model takes the target part class vocabulary, and uses Stable Diffusion, SAM and DINOv2 to get “prototype” embeddings for all classes. Then, it uses a class hierarchy graph (called HPCGraph) and the generated prototype embeddings to segment an image in a top-down manner: first the object-level classes, then the part-level classes, then the subparts, etc. When segmenting the parts and subparts, the model applies a Boundary-Aware Refinement (BAR) module to refine initial segmentation predictions at boundaries between parts that are structurally connected (e.g., “dog-head” and “dog-neck”), as the paper observes that this is where many existing OVPS models struggle. Through experiments, PBAPS is shown to be more effective than existing OVPS models, achieving new state-of-the-art results. Ablation experiments show that the use of the HPCGraph for hierarchical segmentation and the BAR module are essential in achieving this performance.

**Questions:**

My initial rating of this paper is “Borderline Reject”, but I am leaning towards “Reject”. Although the presented method obtains new state-of-the-art results for OVPS, I believe the paper contains too many flaws to be accepted in its current state. Importantly, the paper does not acknowledge the similarity of aspects of the presented method with existing work, making it difficult to judge how significant the contributions are. Moreover, the clarity of several parts of the paper is poor, and an important limitation is not addressed.

In the rebuttal, I would encourage the authors to attempt to address my concerns as stated in the “weaknesses” section, and specifically to (a) explain how the proposed method compares to existing work and (b) clarify the operation of the method.

**Ethical Concerns:**

["NO or VERY MINOR ethics concerns only"]

**Final Justification:**

Based on the different reviews, rebuttals, and the discussion with the authors, I upgrade my rating to 4 (borderline accept), but I remain concerned about some aspects of the paper.

In my initial review, I was mostly concerned about (a) improper attribution to existing work regarding both the prototype generation and hierarchical segmentation, (b) missing conceptual comparisons with existing methods, (c) the readability of the paper, (d) missing explanations, and (e) the limited scalability of the HPCGraph.

In the rebuttal, the authors have addressed most of these concerns by providing additional explanations and promising to revise the paper. Therefore, I am inclined to upgrade my rating. However, this upgrade is contingent on two things:

1.	I remain concerned about the scalability of the HPCGraph, as it will require much manual effort to scale it to a much larger vocabulary with nontrivial object-part hierarchies. However, the authors acknowledge this limitation in the rebuttal, and they explain that there are manners to mitigate it. As such, I am okay with this paper being accepted as long as the authors clearly acknowledge this limitation in the paper, and explain how it can be mitigated.
2.	As highlighted in the review and during the discussion: if the paper is accepted, then the text of the paper needs to be updated significantly to improve readability and understandability. However, there is no way to verify or guarantee that the authors will truly resolve all the issues in the text in a revision, as it is not possible to provide an updated manuscript during the rebuttal period. However, I am willing to give the authors the benefit of the doubt on this matter, because they have provided an example of what the updated abstract will look like and explained how the other parts of the paper will be revised.

Regarding the reviews by the other reviewers, I think the authors have properly addressed most of the identified concerns.

**Limitations:**

The paper lists some limitations of the work in the appendix, but does not mention the limitation of scalability. See weakness 7 in the 'Strengths and Weaknesses' part for more details.

**Quality:**

2

**Strengths And Weaknesses:**

**Strengths**

1.	The presented model, PBAPS, is shown to be effective. On various OVPS benchmarks, it significantly outperforms existing models. This demonstrates the value of the presented model.
2.	The proposed HPCGraph and BAR module are shown to be effective and essential to achieve the state-of-the-art performance that PBAPS obtains, through ablation experiments in Tab. 3. This validates the design of the proposed components.
3.	The qualitative results shown in Fig. 5 provide relevant insights into the actual quality of the predictions by the different models. They show that PBAPS is better able to identify the correct boundaries between parts of an object, which was one of the objectives of this work.

**Weaknesses**

1.	The overall method – i.e., the use of Stable Diffusion, SAM, and DINOv2 to obtain class prototypes that can be used for open-vocabulary segmentation – is largely based on existing method RIM [a], but the paper does not acknowledge this in the paper nor provides proper attribution to this work. The paper does cite RIM as one of several open-vocabulary segmentation models in the related work, and compares to it experimentally in Tab. 2, but it does not mention that the overall method is largely based on it. The submitted paper should mention that the overall operation of PBAPS is largely based on RIM, and explain how it differs from this existing work. Besides being good scientific practice, this enables readers to properly judge the significance of the contributions of the work, which is not possible in the current version.
2.	The concept of top-down, hierarchical object-to-part segmentation – called “progressive” segmentation in this work – is not new, e.g., it is also done by ViRReq [b] and TAPPS [c]. The paper should acknowledge these works, and explain how the proposed top-down segmentation approach differs from these existing approaches, to clarify what the significance is of the presented contribution. This is currently not done.
3.	In general, the paper does not clearly describe how the proposed method compares to existing OVPS methods. The paper does list several existing OVPS methods in L72-L87, and briefly summarizes how they work, but it does not contrast the overall operation of PBAPS to that of existing methods. As such, again, the significance of this work’s contributions cannot be judged fully.
4.	The abstract (L4) claims that this work proposes the first training-free OVPS framework. However, in Tab. 2, there are two other methods (OVDiff and RIM) that are also training-free. Although these methods were not specifically designed for OVPS but for open-vocabulary semantic segmentation, they do work for the OVPS task. Therefore, the claim in L4 is not fully justified, and it would be better if it were removed or altered.
5.	Several sections of the paper are hard to understand and follow because they use (technical) terms that have not been defined or explained yet. This limits the readability and clarify of the paper. For example:


    -	L6-L10 of the paper uses many technical terms to describe the method, e.g., “cross-hierarchy matching”, “matching cost divergence”, “deterministic-region context”. However, these are not commonly used terms, but terms that are introduced in this paper, in later sections. As a result, it is not clear what these terms mean, and the abstract is difficult to understand. The abstract would be much clearer if (a) the method were explained at a higher abstraction level or (b) if these terms were explained.

   -	Similarly, L40-L46 of the introduction describes the BAR module, but this description refers to concepts from the overall method that have not been explained yet, e.g., “part prototypes”, “matching cost”, “original part masks”. As a result, this description cannot be properly understood yet. The clarity of the introduction could be improved by first describing the overall operation of the method (currently done in L47-L52) and introducing these concepts, before explaining the operation of the BAR module.

   -	Similarly, L114-L118 refers to “deterministic regions” and “part prototypes”, concepts that have not been explained yet.

6.	Although the rough, overall operation of the method becomes clear when reading the paper, several aspects of the model are not explained in sufficient detail. This limits understandability and reproducibility of the work. For example:

   -	L130-L131 mentions that cross-attention maps are extracted. However, the paper does not explain how these are extracted. The submitted paper cites 3 other works in this sentence, but it is not clear what exact operation is used.
   -	L134 describes that DINOv2 extracts features from masked regions, but it is not clear if masked image is fed through DINOv2, or if the full image is fed, after which only the per-pixel DINOv2 features that belong to the part masks are used to generate the visual prototype.
   -	L135 makes it seem like there is only one visual prototype per class $c$, but L208 and L522 suggest that there are multiple subcategory prototypes per class. This is inconsistent. Are there multiple prototypes per class, or just a single one? This should be clarified.
   -	In the cost divergence map $D_ {A,B}$ (L165-L166), pixels that have a low matching cost with *both* part $A$ and $B$ will also have a low divergence. This could happen at pixels outside of the mask of the parent class, where it is quite certain that both part classes are not present. In this case, there is clearly no semantic ambiguity, but there is still a low cost divergence. Hence, the statement that “low divergence values indicate regions where both parts exhibit similar matching costs, leading to semantic ambiguity” is not fully true. I assume, therefore, that there is some masking with the parent segmentation mask or initial part segmentation mask, but this is not mentioned or explained. This masking is only done for the deterministic regions $A_ {\textrm{det}}$ and $B_ {\textrm{det}}$ (Eq. 3), but not for the ambiguous regions $U$. This should be clarified.
   -	L204 mentions that “hundreds” of images are generated for each part class, but this is quite vague. Without knowing the exact number, the work is not reproducible.

7.	A limitation of the method, which is not acknowledged in the paper, is that the method does not scale well to a larger vocabulary. If the objective is to do segmentation with a truly 'open' vocabulary, then we would like to have a vocabulary of perhaps millions of concepts. CLIP-based (or other text-embedding-based) methods scale to this relatively well, as we would only have to feed the text description of these concepts through the text encoder, and we could use them directly for segmentation. In the presented PBAPS method, however, we would have to feed all these millions of concepts through Stable Diffusion, SAM, and DINOv2, generating hundreds of millions of images to obtain the visual prototypes, which is rather inefficient. Moreover, we would have to manually construct the hierarchical HPCGraph, which would become intractable at such a scale. I would encourage the authors to acknowledge and reflect on this limitation in the submitted work.

There are some minor additional weaknesses that do not significantly impact my rating:

-	L132 and L167: The symbol $A$ is used for two different purposes, which is confusing.
-	L217: the spaces should be removed, “7. 74” should be “7.74”, etc.
-	L238: “effectivenes” should be “effectiveness”

[a] Wang et al., “Image-to-Image Matching via Foundation Models: A New Perspective for Open-Vocabulary Semantic Segmentation,” CVPR 2024.

[b] Tang et al., “Visual Recognition by Request,” CVPR 2023.

[c] De Geus et al., “Task-aligned Part-aware Panoptic Segmentation through Joint Object-Part Representations,” CVPR 2024.

---

> ### Author Rebuttal · Authors · 2025-07-30
>
> We thank reviewer for the constructive comments. We provide our feedbacks as follows.
>
> ---
> >**W1-W3: Comparison with prior works.**
>
> **A1:** We agree that distinguishing our work from prior methods is essential for clarifying its contributions. In response, we provide the following explanation:
>
> - DiffuMask [45] uses attention maps from Stable Diffusion to obtain category masks for generating semantic segmentation datasets. OVDiff [46] combines diffusion models, segmenters, and feature extraction models to achieve OVSS. RIM [38], building on OVDiff, introduces a stronger feature encoder (DINOv2) and segmenter (SAM), and then conducts relation-aware matching. We highly recognize the contributions of these works, among which the visual prototype construction module provides an important foundation for subsequent research. Our paper follows this technical route, but **our innovation does not lie in prototype construction. We focus on the problem of "low segmentation accuracy for structurally connected part boundaries" in OVPS.** Guided by HPCGraph, hierarchical segmentation is performed, and the BAR module improves boundary precision via feature optimization and dynamic prototype mechanisms. This is also the key reason why our method outperforms OVDiff, RIM, and other methods on the three datasets (as shown in Table 2).
>
> - The progressive segmentation guided by HPCGraph also differs significantly from the top-down fine-grained segmentation in ViRReq [24] and TAPPS [62]. Firstly, when calculating the matching score between pixel features and visual prototypes (L145-147), we fully utilize object-part and part-part relationships. The matching cost between a pixel feature f and a node $v ∈ V$ is defined as the maximum cosine similarity within its dominance set $D(v)$ = {$v$} $\cup$ { descendant nodes of $v$ }. **This combines the current node with all its child nodes into a joint semantic space, thereby fully leveraging hierarchical context to enhance discriminative power.** In addition, the adjacency edges in HPCGraph representing the structural connection between two parts (L141-144). These adjacency edges guide BAR to refine the boundaries of structurally connected parts.
>
> - It should be emphasized that we have never intentionally avoided referencing existing works, nor have we claimed visual prototype construction as our innovative contribution. In Section 2 (Related Work), we provided a brief overview of the aforementioned related works. In Table 2, our method was also fairly compared with these works. Meanwhile, to avoid misleading readers, Section 4.1 only briefly describes the prototype generation process and cites relevant works. Due to the space constraints of the main text, we did not elaborate on the differences between our method and existing works. In the subsequent revision of the paper, we will add a separate section to this end, helping readers more intuitively understand the core contributions and research significance of our work.
>
> >**W4: The statement "propose the first training-free OVPS framework" in the abstract may cause ambiguity.**
>
> **A2:** Thank you for your valuable comment, and we apologize for any confusion caused by the original wording regarding “the first training-free OVPS framework.” We provide the following clarification:
> - To avoid any potential ambiguity, we will revise the original abstract statement to: “We propose a novel and effective training-free framework specifically designed for OVPS...” This revision more accurately convey the positioning and contributions of this work.
> - Although OVDiff and RIM can indeed be applied to OVPS, they are not inherently designed for part-level segmentation tasks. Specifically, these methods focus on object-level OVSS, and their usage in OVPS represents only an indirect adaptation. **They do not address OVPS-specific challenges such as structural dependencies between parts and boundary ambiguity.**
> - In contrast, our method PBAPS is a novel and effective training-free framework explicitly designed for the OVPS task. It targets the core challenge of “low segmentation accuracy at structurally connected part boundaries.” Under the guidance of HPCGraph, PBAPS performs hierarchical segmentation and refines boundaries via the BAR module. As shown in Table 2, under the same prototype generation setup, PBAPS outperforms the state-of-the-art method RIM by **7.3%, 18.1%**, and **8.4%** on Pascal-Part-116, ADE20K-Part-234, and PartImageNet, respectively.
>
>
> >**W5: Readability of the paper.**
>
> **A3:** Thank you for your suggestions regarding the readability of the paper. We will make the following specific revisions:
> - For non-generic terms such as "cross-hierarchy matching" and "deterministic region" that appear early in the abstract and introduction, we will add brief, context-rich explanations when they first appear  (e.g., explaining how "deterministic regions" serve as reliable context for refining ambiguous boundaries), ensuring readers grasp their core meaning upfront.
> - We will restructure the abstract and introduction to first outline the overall goal of OVPS and highlight the key challenge of structurally connected part boundaries. The PBAPS framework and its core modules (HPCGraph, BAR) will then be introduced progressively in a logical flow, with technical details deferred until the basic concepts are established. Additionally, we will simplify overly technical language to better emphasize the motivation, main contributions, and high-level workflow, thereby improving accessibility for readers without a specialized background.
>
>
> >**W6: Some operations in the paper lack sufficiently detailed explanations.**
>
> **A4:** Thank you for pointing out that some operations in this paper lack detailed explanations. In response, we provide the following supplementary notes:
> - Regarding the extraction of cross-attention maps in L130–131. To avoid misleading readers into thinking this is an innovation of our work, we only give a brief description of this part in the main text (L130–131) and provide more details in Appendix B.1. We further supplement as follows: we follow the prior works such as DiffuMask [45], OVDiff [46], and RIM [38], obtaining robust localization signals through cross-timestep and cross-layer aggregation. Cross-attention maps are extracted from all 50 diffusion timesteps, and noise from a single step is suppressed through an averaging operation. The attention maps come from the 16×16 resolution cross-attention layers generated during the downsampling process of the U-Net, as well as the 32×32 and 64×64 resolution cross-attention layers generated during the upsampling process. Finally, attention maps of different resolutions are scaled to 512×512 through interpolation and weighted averaged to generate the final semantic localization heatmap.
> - Regarding the feature extraction for masked regions in L134. We use the masked image as input to DINOv2 for pixel feature extraction, instead of extracting features from the entire image first and then applying masking. This pre-masking operation enhances the accuracy of prototype construction.
> - Regarding the inconsistent descriptions of the number of visual prototypes (L135, L208, L522). Each part category corresponds to multiple subcategory prototypes, which is the configuration of our method. However, in L135, for the sake of narrative simplicity and to help readers understand, we described the visual prototype as "one", and this wording may have caused misunderstandings. We apologize for this.
> - Regarding the calculation range of the cost divergence map D (L165–166). Your understanding is correct. The cost divergence map is calculated only within the parent mask region. When constructing the cost divergence map, we only consider pixels that belong to the upper-level part segmentation mask (parent mask) to avoid misjudgment caused by irrelevant regions.
> - Regarding the number of synthetic images (L204). For most part categories, we generated 120 prototype images; only for a few categories with poor generation results, an additional 30 images were generated to improve prototype quality. Since we cannot add images, PDFs, or links during this NeurIPS rebuttal stage, we will provide a complete list of image quantity configurations for all categories in subsequent revisions of the paper and make the corresponding image data publicly available.
>
>
> >**W7: The scalability of HPCGraph.**
>
> **A5:** Due to the space constraints of the rebuttal, we kindly refer you to our detailed response to Reviewer 76j9’s Weakness 3, where we provide an in-depth explanation of the scalability of HPCGraph, including its construction principles, reuse mechanism across structurally similar categories (group 158 classes into 11 superclasses, topology count drops by **93 %** while PBAPS still leads RIM by **8.4 %**), and dynamic graph generation mechanism (**automatically add adjacency edges for highly relevant parts based on the spatial adjacency and visual feature similarity between part response regions**). The comparative results between dynamic and static graphs are as follows:
> | HPCGraph | **Pascal-Part-116**  | **ADE20K-Part-234** | **PartImageNet**|
> |:-|:-:|:-:|:-:|
> |  | mIoU&ensp;&emsp;bIoU | mIoU&ensp;&emsp;bIoU| mIoU&ensp;&emsp;bIoU |
> | Dynamic | 44.63&ensp;&emsp;33.12 |23.87&ensp;&emsp;15.63| 41.66&ensp;&emsp;28.72 |
> | Static | **46.35**&ensp;&emsp;**34.46** |**24.70**&ensp;&emsp;**16.41** | **42.61**&ensp;&emsp;**29.31** |
>
> ---
>
>
> [24] Visual recognition by request (CVPR 2023)
>
> [38] Image-to-Image Matching via Foundation Models: A New Perspective for Open-Vocabulary Semantic Segmentation (CVPR 2024)
>
> [45] Diffumask: Synthesizing391 images with pixel-level annotations for semantic segmentation using diffusion models (ICCV 2023)
>
> [46] OVDiff: Diffusion Models for Open-Vocabulary Segmentation (ECCV 2024)
>
> [62] Task-aligned Part-aware Panoptic Segmentation through Joint Object-Part Representations (CVPR 2024)

---

> > ### Comment · Reviewer_4fTr · 2025-08-02
> > **Response to Authors**
> >
> > Thank you for writing a rebuttal to my review and the author reviews, and for providing additional explanations and clarifications. I have carefully read the rebuttals and I will take them into account during the reviewer-AC discussion and when determining my final recommendation.
> >
> > Currently, I have a few remaining questions:
> >
> > 1. **Regarding readability**: The rebuttal states that the abstract and introduction will be restructured and that non-generic terms will be explained when they first appear, e.g., early in the abstract. What will the revised abstract look like exactly? Could you please provide a revised version of the text?
> > 2. **Regarding scalability**:
> >    * The rebuttal discusses the scalability of the construction of the HPCGraph, but doesn't reflect upon the following statement of my review:
> >       > In the presented PBAPS method, however, we would have to feed all these millions of concepts through Stable Diffusion, SAM, and DINOv2, generating hundreds of millions of images to obtain the visual prototypes, which is rather inefficient.
> >
> >       Could you please reflect upon this? Do you agree/acknowledge that this is a fundamental limitation? Even if semantic categories can share part hierarchies in the HPCGraph (as done for PartImageNet), a true open-vocabulary model would require the generation of millions of images to obtain the visual prototypes, which is inefficient.
> >    * The rebuttal to reviewers *Sy1A*, *Zzg*, and *76j9* states:
> >       > HPCGraph is constructed based on commonsense structural knowledge [..] and does not rely on any manual annotations.
> >
> >       However, it is not at all clear to my why it does not rely on manual annotations. In the end, the part categories, object-part relationships, and part-part relationships that are required for the HPCGraph need to be defined manually, right? Someone has to consider all the different concepts and determine what constitutes a 'part' and a 'subpart', and assess which parts are structurally connected. If this has to be done for a truly open vocabulary (i.e., millions of concepts), this would require a lot of manual labor (even when aided by an LLM), meaning that the method does not scale well.
> >    * The rebuttal to reviewers *Sy1A*, *Zzg*, and *76j9* states that visual prototypes could be reused, e.g., 'horse-leg' could be applied to 'donkey-leg'. Are there empirical results that demonstrate that this can be done, without the necessity of generating additional images for 'donkey-leg' to update the prototypes? In the case of PartImageNet, where 11 superclasses use the same topological graph, were the visual prototypes of part classes reused, e.g., by using the 'horse-leg' prototype also for 'donkey-leg'? Or were the prototypes for the shared 'horse/donkey/zebra/mule-leg' class obtained by generating images for all the individual part concepts, and then aggregating them into a set of joint visual prototypes? In case that the former is not true, then how do you know that visual prototypes can be reused?
> >
> > Thanks in advance for your response.

---

> ### Author Response · Authors · 2025-08-03
>
> **(1/3)**
>
> Thank you sincerely for your timely feedback. We provide our feedbacks as follows.
>
> ---
> >**Q1: What will the revised abstract look like exactly? Could you please provide a revised version of the text?**
>
> **A1:** Thank you for your suggestions to help refine our paper. Following your advice, only minor revisions are needed to achieve clearer readability. For example, we replaced the original term "matching cost divergence" with "measuring classification uncertainty", and "deterministic-region context" with “high-confidence context"—phrases that more intuitively convey the underlying ideas without assuming prior familiarity with our technical design. The revised abstract is as follows:
>
> *Open-vocabulary part segmentation (OVPS) struggles with structurally connected boundaries due to the conflict between continuous image features and discrete classification. To address this, we propose PBAPS, a novel training-free framework specifically designed for OVPS. PBAPS leverages structural knowledge of object-part relationships to guide a progressive segmentation  from objects to fine-grained parts. To further improve accuracy at challenging boundaries, we introduce a Boundary-Aware Refinement (BAR) module that identifies ambiguous boundary regions by **measuring classification uncertainty** (quantified via the difference in classification scores between adjacent parts), enhances their features using **high-confidence context** (through fusion with features from reliably classified pixels), and adaptively refines part prototypes to better align with the specific image. Experiments on Pascal-Part-116, ADE20K-Part-234, PartImageNet demonstrate that PBAPS significantly outperforms state-of-the-art methods, achieving 46.35% mIoU and 34.46% bIoU on Pascal-Part-116.*
>
> >**Q2: We would have to feed all these millions of concepts through Stable Diffusion, SAM, and DINOv2, generating hundreds of millions of images to obtain the visual prototypes, which is rather inefficient. Even if semantic categories can share part hierarchies in the HPCGraph (as done for PartImageNet), a true open-vocabulary model would require the generation of millions of images to obtain the visual prototypes, which is inefficient.**
>
> **A2:** We fully agree with your point: when scaling to millions of semantic concepts, the process of generating visual prototypes indeed faces efficiency limitations. However, it should be clarified that such efficiency limitations are relative to the extreme scenario where "no reuse mechanism is employed, requiring the generation of hundreds of millions of images." Regarding your statement that "Even if..., a true open-vocabulary model would require the generation of millions of images to obtain the visual prototypes, which is inefficient." , we have verified the feasibility of efficiency through actual generation tests: **using 8 A6000 GPUs for parallel generation, we have accumulated 181K images over 21 hours**. Based on this, it is estimated that generating one million images would take no more than 5 days, an efficiency acceptable in practical applications. We further supplement from the perspectives of **technical background** and **work positioning**:
> - First, to the best of our knowledge, no semantic segmentation method can truly and efficiently handle millions of concepts. The core obstacle lies in the requirement for massive densely annotated data to train such models. The annotation costs alone far exceed practical feasibility, and the model also faces huge challenges in representing and generalizing such a large semantic space. Thus, the core goal of open-vocabulary segmentation is **not to "cover infinite categories," but to break free from reliance on large-scale pixel-level annotations**, exploring universal segmentation solutions for low-resource, weakly supervised, or zero-shot scenarios. In this context, our method uses visual prototype matching for label assignment, leveraging the validated pipeline to obtain visual prototypes, which aligns with the fundamental intent of open-vocabulary tasks.
> - Second, the core contribution of our work is not to "expand category coverage," but to address segmentation errors caused by "ambiguous boundaries and structural coupling between parts" under the open-vocabulary setting. It should be emphasized that PBAPS itself is agnostic to the source of prototypes, offering high plug-and-play flexibility and universality. **PBAPS does not rely on a specific prototype generation pipeline**. We can directly leverage large-scale image-text pair data from the internet, extract candidate regions (via SAM), and then use CLIP to match regions with text, thereby mining highly diverse and naturally distributed part prototypes in the real world.

---

> ### Author Response · Authors · 2025-08-03
>
> **(2/3)**
>
> >**Q3: HPCGraph is constructed based on commonsense structural knowledge [..] and does not rely on any manual annotations. It is not at all clear to my why it does not rely on manual annotations. If this has to be done for a truly open vocabulary (i.e., millions of concepts), this would require a lot of manual labor.**
>
> **A3:** We apologize for the ambiguity. By "HPCGraph does not rely on manual annotations," we specifically mean it avoids costly dense visual annotations like pixel-level masks or bounding boxes used in traditional supervised training.
>
> **The construction of HPCGraph can be based on existing general knowledge bases (such as WordNet, ConceptNet, etc.)**, which widely contain semantic relationships between objects and parts like "part-of" and "has-part". Compared with the traditional method that relies on pixel-level annotation, this kind of structural knowledge definition has controllable workload and strong cross-category reusability—with superclass merging and LLM-assisted parsing, manual effort is further reduced.
>
> Regarding the effectiveness of the topology reuse mechanism, it has been verified on PartImageNet in the paper: after dividing 158 categories into 11 superclasses, the number of topological structures is reduced by 93%, while PBAPS improves upon RIM by 8.4% relatively (Table 2), indicating that topological structures have good shareability. **We further supplement the judgment method for topology reuse**: based on the hierarchical taxonomy provided by WordNet, we first query the hypernym chain of a target category (e.g., “tiger” is a hyponym of “quadruped”) and then combine this with structural features such as limb count and part connectivity to identify semantically close and structurally similar categories (e.g., “tiger,” “leopard,” and “golden retriever” all share a “head-torso-four-limb” configuration). These are grouped into the same superclass (e.g., “quadruped animals”), which is then assigned a unified part topology.
>
> To verify the scalability of this mechanism, we follow prior work[63] and extract 10,000 concepts from the CC3M dataset. For each concept, we obtain its WordNet hypernym chain (e.g., "Persian cat" → "cat" → "mammal" → "quadruped") and cluster concepts with semantic distance ≤2 (e.g., "dog," "wolf," and "fox" into the superclass "canine quadrupeds") based on convergence in their upper-level hierarchy. In this way, the **10,000 concepts are grouped into 40 superclasses, each corresponding to a shared topology**. This process not only ensures the validity of topology reuse within each superclass but also significantly reduces the number of topologies required under large-scale settings, validating the feasibility and efficiency of our topology reuse mechanism.
>
> [63] Image-Text Co-Decomposition for Text-Supervised Semantic Segmentation (CVPR 2024)
>
> [64] Conceptual Captions: A Cleaned, Hypernymed, Image Alt-text Dataset For Automatic Image Captioning (ACL 2018)

---

> ### Author Response · Authors · 2025-08-03
>
> **(3/3)**
>
> >**Q4: The rebuttal states that visual prototypes could be reused, e.g., 'horse-leg' could be applied to 'donkey-leg'. Are there empirical results that demonstrate that this can be done? How do you know that visual prototypes can be reused?**
>
> **A4:** Since part prototypes directly participate in the calculation of classification scores, the conditions for reuse are more stringent than those for part-topology reuse. Even within the same superclass, it is not appropriate to naively apply a single set of part prototypes to all categories. To address this, we **employ LLMs to analyze local structural and textural similarities between categories and determine whether reuse is appropriate**.
>
> We designed experiment on the PartImageNet dataset to verify the feasibility of part prototype reuse. PartImageNet contains 40 object categories, which can be divided into 11 superclasses based on their morphological and structural characteristics. All objects within a superclass share the same part topology. It should be noted that due to the significant differences in overall appearance between different object categories (e.g., "goldfish" and "killer whale"), we still need to generate separate visual prototypes for each object to ensure the accuracy of object-level segmentation. At the part level, we explored the possibility of reusing prototypes across objects.
>
> For each superclass, we selected a most representative category (e.g., "tiger" for the "Quadruped" superclass) and generated complete part visual prototypes for it (including tiger head, tiger torso, etc.). Next, through LLM analysis and screening: objects with significant differences in local structure or texture from the representative category (e.g., "giant panda" vs. "tiger") are excluded from reuse candidates to avoid introducing errors. After screening, the part segmentation of the remaining objects in the superclass directly reuses the part prototypes of the representative category without the need to generate new prototypes. The specific representative categories and target categories are shown in the table below:
>
> | Superclass &emsp;&emsp;| Representative Category |&emsp;&emsp;&emsp;&emsp;Target Categories |
> |:-|:-|:-|
> |Quadruped|tiger|leopard, golden retriever|
> |Snake|green mamba|Indian cobra|
> |Reptile|green lizard|Komodo dragon, American alligator|
> |Boat|yawl|pirate, schooner|
> |Fish|goldfish|barracouta, killer whale, tench|
> |Bird|albatross|bald eagle|
> |Car|garbage truck|minibus,ambulance,school bus|
> |Bicycle|mountain bike|moped, moto rscooter|
> |Biped|gorilla|orangutan, chimpanzee|
> |Bottle|water bottle|beer bottle, wine bottle|
> |Aeroplane|warplane|airliner|
>
> We compared the performance of models with and without the part prototype reuse strategy, and the results are shown in the table below. Although the reuse of part prototypes leads to a certain degree of degradation in segmentation performance, the overall performance remains well usable. In scenarios with large-scale categories, this strategy significantly reduces the cost of generating visual prototypes while maintaining good segmentation results. In summary, **under the premise of high consistency in local structure and texture within a superclass, the reuse of visual prototypes is feasible**, eliminating the need to generate entirely new part prototypes for each object category.
>
> |Reuse Visual Prototypes|Representative Category|Target Category|&emsp;Average&emsp;|
> |:-:|:-:|:-:|:-:|
> ||miou&emsp;&emsp;biou|miou&emsp;&emsp;biou|miou&emsp;&emsp;biou|
> | Yes |43.62&emsp;&emsp;30.43|39.14&emsp;&emsp;26.82|40.68&emsp;&emsp;28.06|
> |No|43.72&emsp;&emsp;30.45|42.52&emsp;&emsp;28.88|42.93&emsp;&emsp;29.42|
>
> ---
>
> We once again extend our sincere respect to you. Regarding the responses during the Rebuttal stage and the supplementary content provided this time, if you still have any questions or need further clarification on any aspects, please do not hesitate to let us know. We will be happy to provide explanations for you.

---

> > ### Comment · Reviewer_4fTr · 2025-08-04
> > **Response to Authors**
> >
> > Thank you very much for the additional explanations and results. I will take these into consideration when formulating my final recommendation.

---

> > > ### Author Response · Authors · 2025-08-06
> > >
> > > Dear Reviewer,
> > >
> > > Thank you again for your thoughtful and constructive feedback on our submission. We truly appreciate the time and effort you have devoted to the review process.
> > >
> > > With only a limited time budget left in the discussion period, we would like to kindly confirm whether our previous responses have fully addressed your concerns. Your insights have been invaluable, and we’ve made every effort to respond with clarity and precision. If anything remains unclear or if further clarification would be helpful, we would be more than happy to continue the discussion.
> > >
> > > We would be sincerely grateful if you could explicitly let us know whether our explanations have resolved your concerns. Thank you once again for your time and consideration.
> > >
> > > Best regards,
> > >
> > > The authors of Paper 8092

---

> > > > ### Comment · Reviewer_4fTr · 2025-08-07
> > > > **Response to Authors**
> > > >
> > > > With the additional explanations and clarifications from the rebuttal, several - but not all -  of my concerns have been addressed.
> > > >
> > > > Regarding the originally identified weaknesses:
> > > > * **1 - prototype generation based on existing work without proper attribution:** I understand that the prototype generation method is not the main contribution, but I still think the paper should better acknowledge the similarity with existing work. The rebuttal states that Sec. 4.1 cites the relevant works, but in L131 the paper only cites works when talking about using Stable Diffusion to generate synthetic images and localize parts. It doesn't state that the entire, overall prototype generation method is based on RIM [38]. This should be done to make it clear which technical components are newly proposed and which are taken from existing work.
> > > > * **2 - top-down (hierarchical) segmentation is not new:** I understand that PBAPS additionally conducts cross-hierarchy matching, and that ViRReq and TAPPS do not do this. Still, given the fact that ViRReq and TAPPS do already conduct hierarchical segmentation, I believe the paper should clearly indicate that hierarchical segmentation is not new, and that this cannot/should not be claimed as a contribution.
> > > > * **3 - explanation of comparison to existing OVPS methods:** This concern has partially been addressed by better comparisons to DiffuMask, RIM, and OVDiff. However, the paper still doesn't explain why PBAPS is better than existing OVPS methods like PartCLIPSeg, OIParts, PartCATSeg, etc. L72-L87 lists some of these methods and briefly explains their operation, but the paper does not explain how they compare to PBAPS, and why PBAPS obtains a performance that is so much better.
> > > > * **4 - statement about "first training-free OVPS framework:** This concern has been addressed with the adjusted abstract provided in the rebuttal.
> > > > * **5 - readability of the paper:** This concern has been mostly addressed. The adjusted abstract provided in the rebuttal resolves my concerns, and I assume that similar changes will be made in the rest of the paper, according to the promises made by the authors in the rebuttal.
> > > > * **6 - missing explanations:** This concern has been addressed with the additional explanations in the rebuttal.
> > > > * **7 - scalability of HPCGraph:** I still believe that the scalability of the HPCGraph is limited, as it requires manual construction and would have to grow to a very large size to accommodate a truly open vocabulary. However, I acknowledge that there are several ways to mitigate this limitation, e.g., by reusing the part topology for several concepts, reusing visual prototypes, and deploying LLMs or WordNet to aid the manual construction of the graph. As such, if the authors promise to acknowledge the limitation of scalability in the paper, and to discuss the ways to mitigate this, then this concern is mostly addressed.
> > > >
> > > > Given that several of my concerns have been addressed, I am considering updating my score, but I wish to first discuss this paper among the reviewers, in the *reviewer-AC discussion* that will follow after the *reviewer-author discussion*.

---

> > > > > ### Author Response · Authors · 2025-08-07
> > > > >
> > > > > We sincerely appreciate the in-depth feedback you provided during the discussion phase. Regarding your concerns, we respond to each item earnestly and commit to supplementing and improving them one by one in the paper revision:
> > > > >
> > > > > ---
> > > > > >**W1: Insufficient attribution of prototype generation to existing work**
> > > > >
> > > > > We fully agree with your suggestion and will explicitly emphasize in the paper that the visual prototype generation scheme is based on the technical route of RIM. For example, the description of RIM in the Related Work (L65) will be revised to: "RIM [38] employs image-to-image matching for training-free segmentation, constructing visual references and enhancing robustness via a relation-aware ranking distribution strategy. **Our method generates part prototypes based on RIM process of building visual references (see Sec. 4.1 for details)."** Additionally, the beginning of Section 4.1 will clearly state that this process is derived from RIM, helping readers better distinguish the innovations of this paper from existing technical foundations.
> > > > >
> > > > > >**W2: Clarifying that hierarchical segmentation is not a new technique**
> > > > >
> > > > > We fully agree with your view. We will supplement the related work TAPPS in the Related Work section (L77) and clearly state: **"Hierarchical segmentation is not a novel concept—ViRReq and TAPPS have previously implemented top-down object-to-part segmentation**. In contrast, the hierarchical reasoning in this paper additionally introduces a cross-hierarchy matching mechanism, making full use of hierarchical context to enhance discriminative power, which is the core difference from the aforementioned works." We will strictly avoid presenting "hierarchical segmentation" as a contribution of this paper, but instead focus on the extensions and optimizations based on it.
> > > > >
> > > > > >**W3: Insufficient comparison with existing OVPS methods**
> > > > >
> > > > > Thank you for pointing out this deficiency. Although Table 2 has demonstrated the superiority of PBAPS, the paper does not fully explain why PBAPS outperforms existing OVPS methods. **We will revise and supplement L72-L87 to compare existing OVPS methods with our PBAPS**, for example: "PartCLIPSeg adopts a single-stage architecture and improves part recognition ability through multi-granularity image-text alignment, but it lacks explicit spatial structure constraints, which easily leads to the phenomenon of 'out-of-object activation' and has insufficient ability to segment the boundaries of structurally connected parts. PBAPS effectively alleviates the above problems through the hierarchical segmentation strategy guided by HPCGraph, and combines the uncertainty modeling and feature refinement of the BAR module to significantly improve the segmentation accuracy of complex structural regions. PartCATSeg enhances semantic discrimination by constructing part-aware text embeddings combined with contrastive training, but it relies on training data and has weak generalization ability when dealing with rare categories or structurally complex objects. Meanwhile, it lacks the ability to model structural dependencies between parts. In contrast, PBAPS builds visual prototypes based on RIM, and under the premise of training-free, realizes cross-hierarchy matching and structural prior modeling through HPCGraph, combined with the BAR module to effectively solve core challenges such as semantic confusion and boundary ambiguity."
> > > > >
> > > > > >**W4-W6: Readability, accuracy, and terminology explanation**
> > > > >
> > > > > Thank you for your trust and support for our work. We will strictly follow our commitment to optimize the expression in the abstract, introduction, and method sections. For non-general terms, we will provide clear definitions when they first appear and supplement their roles and meanings in the context. At the same time, we will improve all operational details that have not been fully explained, and supplement one by one the unclear expressions or insufficient explanations in the paper.
> > > > >
> > > > > >**W7: Scalability of HPCGraph**
> > > > >
> > > > > We understand your concern about the scalability of HPCGraph and will clearly point out the limitations of this mechanism in large-scale open vocabulary scenarios in the paper. Meanwhile, we will also elaborate on several feasible mitigation strategies, including: (1) topology reuse; (2) visual prototype reuse; (3) using WordNet and LLM to assist in building HPCGraph. These strategies can significantly improve the practicality and scalability of PBAPS.
> > > > >
> > > > > ---
> > > > >
> > > > > Thank you again for the time and energy invested in the review process, as well as your recognition and tolerance of this work. We solemnly commit to supplementing and improving the above contents one by one in the subsequent paper revision to further enhance the clarity, scientificity, and completeness of the paper. If you have other suggestions, we are also very willing to continue communicating and improving.

---

### Official Review · Reviewer_76j9 · 2025-06-29

**Clarity:** 3
**Significance:** 3
**Originality:** 3
**Rating:** 5
**Confidence:** 3

**Summary:**

This paper addresses a critical challenge in Open Vocabulary Part Segmentation (OVPS): the inaccurate delineation of boundaries between structurally connected parts. To solve this, the authors propose PBAPS, a novel training free framework capable of performing OVPS without any additional learning.

PBAPS consists of three main components:

- HPCGraph, which models the hierarchical relationships between objects and their parts

- BAR (Boundary Aware Refinement), a module that detects ambiguous boundary regions and uses surrounding context to enhance feature representations and adapt visual prototypes

- A training free framework that leverages pre-trained models such as Stable Diffusion, SAM, and DINOv2 to generate visual prototypes and perform segmentation purely through inference

The proposed method achieves state-of-the-art performance on three benchmarks: Pascal Part 116, ADE20K Part 234, and PartImageNet. It shows significant improvement over existing methods, especially in accurately segmenting structurally connected part boundaries.

**Questions:**

- In the Cross-Model Prototype Induction stage, the visual prototypes are entirely dependent on the localization capability of Stable Diffusion. However, Stable Diffusion does not always highlight regions that accurately correspond to the given input prompt. What is the likelihood that the resulting visual prototype actually represents the region corresponding to the prompt?
- Additionally, the cross-attention maps in Stable Diffusion vary significantly depending on the timestep and the layer of the U-Net. From which timestep and which specific layer of the U-Net do you extract the attention maps for localization purposes?

**Ethical Concerns:**

["NO or VERY MINOR ethics concerns only"]

**Final Justification:**

The paper identifies the root cause of segmentation errors at structural boundaries and introduces PBAPS, the first training-free OVPS framework with strong zero-shot generalization. Since the authors have addressed my concern well, I am raising my rating.

**Limitations:**

Yes

**Quality:**

3

**Strengths And Weaknesses:**

Strengths
- The paper clearly identifies the fundamental cause of segmentation errors at structurally connected part boundaries, focusing on the intrinsic conflict between continuous visual features and discrete classification mechanisms, and proposes a solution to address this issue.

- PBAPS is the first training-free framework in the OVPS field, enabling segmentation without any additional training. This approach demonstrates strong zero-shot generalization to unseen part classes.

- The method achieves state-of-the-art mIoU and bIoU scores on three benchmarks (Pascal-Part-116, ADE20K-Part-234, and PartImageNet) surpassing both supervised and existing training-free methods by a significant margin.

Weakness
- Dependence on foundation models: Despite the advantage of being training-free, PBAPS heavily relies on large-scale pre-trained foundation models such as Stable Diffusion, SAM, and DINOv2 for visual prototype generation and feature extraction. As a result, its performance and applicability may be influenced by the quality, availability, and domain compatibility of these models.

- Sensitivity to hyperparameters: The performance of the BAR module is sensitive to hyperparameters such as feature fusion weight, prototype adaptation coefficient, and number of sub-prototypes. Achieving optimal performance requires careful tuning, which can hinder the model’s generalizability and ease of deployment across diverse scenarios.

- Manual construction of HPCGraph: As acknowledged by the authors, the construction of HPCGraph is not an automated process but rather a manual one, relying on predefined object-part hierarchies or external knowledge. This limits the direct applicability of PBAPS to new domains or complex object categories where such structural priors are not readily available.

- Incomplete ablation study: The ablation study for the proposed method is conducted solely on the Pascal-Part-116 dataset. This presents a limitation in assessing the generalizability and robustness of individual components, as it remains unclear whether similar trends would hold across other datasets like ADE20K-Part-234 or PartImageNet.

---

> ### Author Rebuttal · Authors · 2025-07-30
>
> We thank reviewer for the constructive comments. We provide our feedbacks as follows.
>
> ---
> >**W1: Dependence on foundation models.**
>
> **A1:** Regarding PBAPS's dependence on foundation models, we need to clarify: this dependence is not a flaw but rather a best practice in cutting-edge research. The value of this paper lies in **how to leverage the capabilities of foundation models to solve the core challenge of OVPS, namely "low segmentation accuracy for structurally connected part boundaries."**
>
> Foundation models such as Stable Diffusion, SAM, and DINOv2, having been trained on massive amounts of data, possess strong general visual understanding capabilities, which serve as the premise for PBAPS's training-free design. The outputs of foundation models (such as SAM's masks and DINOv2's features) are essentially general visual signals, whereas OVPS requires structured cognitive understanding of part relationships: How to locate part boundaries from diffuse attention maps? How to adapt global prototypes to instance-specific differences in specific images? Our HPCGraph, through structured knowledge constraints, guides the model in hierarchical reasoning; the dynamic prototype mechanism within the BAR addresses the misalignment between foundation model outputs and the actual distribution of parts. These designs are our innovation—not relying on the foundation models, but constructing the paradigm for applying their capabilities. When more powerful foundation models emerge, our method can directly reuse their capabilities without any modifications.
> >**W2: Sensitivity to hyperparameters.**
>
> **A2:** Due to the space constraints of the rebuttal, we kindly refer you to our detailed response to Reviewer Sy1A’s Weakness 2, where we thoroughly address the selection rationale, robustness, and generalizability of our hyperparameter settings across diverse datasets.
>
> >**W3: Manual construction of HPCGraph.**
>
> **A3:** Regarding the scalability of HPCGraph, we provide the following supplementary explanations:
> - HPCGraph is constructed based on commonsense structural knowledge and does not rely on any manual annotations. As such, it incurs low construction cost and exhibits strong transferability and scalability. In large-scale object scenarios, **both the topological structure and visual prototypes can be shared and reused.** For novel object categories with similar structures to existing ones (e.g., “zebra” as a novel category relative to the existing category “horse”), we can directly reuse existing topologies (e.g., “zebra head ↔ zebra neck” correspond to the structure of "horse"). For PartImageNet (L198–200), we divided 158 object categories into 11 superclasses, where each superclass shares a unified topological graph. This strategy reduces the number of topology by **93%** (from 158 to 11) while maintaining high performance—PBAPS outperforms RIM by **8.4%** on PartImageNet (Table 2). When the number of object categories expands to tens of thousands, a large number of them are highly similar in form and structure (such as horses, donkeys, mules, zebras, etc.). We can use LLMs to analyze inter-category structural similarities and achieve cluster-level topology reuse (i.e., categories with similar structures share a set of topologies). Furthermore, part visual prototypes can also be reused (e.g., the prototype for “horse leg” can be applied to “donkey leg”).
> - In contrast to existing methods such as PartCLIPSeg [27] and PartCATSeg [53], which require extensive part-level annotations and retraining when introducing new object categories, our method is training-free and supervision-independent. It supports open-world scalability by simply extending the HPCGraph and adding a small number of visual prototypes.
> - As noted in Appendix A.1, when dealing with objects with irregular structures (e.g., modular furniture), the static graph structure has insufficient adaptability. To address this, a dynamic graph generation mechanism can be considered. The core logic is to **automatically add adjacency edges for highly relevant parts based on the spatial adjacency and visual feature similarity between part response regions**. The following table compares performance using static and dynamic graphs. While static graphs yield better performance in well-structured domains, the dynamic version provides competitive results and significantly enhances flexibility for deployment in unconstrained real-world scenarios.
> | HPCGraph | **Pascal-Part-116**  | **ADE20K-Part-234** | **PartImageNet**|
> |:-|:-:|:-:|:-:|
> |  | mIoU&ensp;&emsp;bIoU | mIoU&ensp;&emsp;bIoU| mIoU&ensp;&emsp;bIoU |
> | Dynamic | 44.63&ensp;&emsp;33.12 |23.87&ensp;&emsp;15.63| 41.66&ensp;&emsp;28.72 |
> | Static | **46.35**&ensp;&emsp;**34.46** |**24.70**&ensp;&emsp;**16.41** | **42.61**&ensp;&emsp;**29.31** |
>
> >**W4: Incomplete ablation study.**
>
> **A4:** In response to your comment, we supplement the ablation experiment results of PBAPS on the ADE20K-Part-234 and PartImageNet datasets. As shown in the table below, on three datasets with significant differences (natural objects, indoor scenes, and cross-domain complex objects), each component significantly improves the model performance (mIoU). This not only verifies the effectiveness of each component design but also demonstrates that our method has good generalizability and robustness in diverse data environments.
>
> | w/ HPCGraph |w/ BAR| **Pascal-Part-116**  | **ADE20K-Part-234** | **PartImageNet**|
> |:-:|:-:|:-:|:-:|:-:|
> | | | mIoU&ensp;&emsp;bIoU | mIoU&ensp;&emsp;bIoU| mIoU&ensp;&emsp;bIoU |
> |  |  |40.74&ensp;&emsp;31.09 |20.59&ensp;&emsp;13.72 | 37.46&ensp;&emsp;25.78 |
> |  | √ |42.95&ensp;&emsp;32.34 |22.14&ensp;&emsp;14.53 | 39.54&ensp;&emsp;27.12 |
> | √ |  |44.08&ensp;&emsp;32.99 |23.62&ensp;&emsp;15.43 | 41.27&ensp;&emsp;28.58 |
> | √ | √ |46.35&ensp;&emsp;34.46 |24.70&ensp;&emsp;16.41 | 42.61&ensp;&emsp;29.31 |
>
> >**Q1: What is the likelihood that the resulting visual prototype actually represents the region corresponding to the prompt?**
>
> **A5:** Thank you for your valuable comment, we provide the following supplementary explanation:
> - It should be emphasized that the approach of constructing visual prototypes using Stable Diffusion + SAM + DINO has been adopted and validated as effective in multiple existing works (e.g., OVDiff [46], RIM [38]). **Our contribution does not lie in the prototype construction, but rather in addressing the challenge of low segmentation accuracy at structurally connected part boundaries in OVPS.** Guided by HPCGraph, hierarchical segmentation is performed, and the BAR module improves boundary precision via feature optimization and dynamic prototype mechanisms. This design is the key reason why our method outperforms OVDiff, RIM, and other methods on the three datasets (Table 2).
> - Regarding whether Stable Diffusion can accurately localize prompt regions, **we conducted a quantitative verification using CLIP cross-modal similarity.** For the synthetic images of each part category, SAM generates target masks based on the attention maps of Stable Diffusion. Subsequently, we used CLIP to calculate the similarity between the masked regions, non-masked regions, and the entire image with the target text prompts, respectively. The results are shown in the table below. The similarity score of the target masked regions is significantly higher than that of the entire image, verifying that Stable Diffusion can accurately focus on the target parts. In addition, the low similarity score of non-masked regions also indirectly confirms the accuracy of the masked regions.
>
> | Source of Image Feature | **Pascal-Part-116** | **ADE20K-Part-234** | **PartImageNet**|
> |:-|:-:|:-:|:-:|
> | Masked Region |0.75±0.04|0.74±0.03|0.72±0.04|
> | Entire Image |0.61±0.05|0.58±0.03|0.56±0.04|
> | Non-masked Region |0.21±0.03|0.20±0.03|0.22±0.02|
> - Furthermore, even if there is a certain deviation in the target masks (e.g., including a small number of non-target regions), leading to ambiguity in the generated part visual prototypes, the "dynamic prototype adaptation" mechanism of the BAR module can effectively alleviate this issue. **By weighted fusion of global prototypes with the features of deterministic regions in the current image, the prototypes are dynamically adapted to the real target parts in the image**, thereby improving segmentation accuracy. The ablation experiment in Table 3 also confirms the effectiveness of the "dynamic prototype adaptation" mechanism.
>
> >**Q2: From which timestep and which specific layer of the U-Net do you extract the attention maps for localization purposes?**
>
> **A6:** As shown in L129-131 and L456-467, we follow the prior works such as DiffuMask [45], OVDiff [46], and RIM [38], obtaining robust localization signals through cross-timestep and cross-layer aggregation. Cross-attention maps are extracted from all **50** diffusion timesteps, and noise from a single step is suppressed through an averaging operation. The attention maps come from the **16×16** resolution cross-attention layers generated during the downsampling process of the U-Net, as well as the **32×32** and **64×64** resolution cross-attention layers generated during the upsampling process. Finally, attention maps of different resolutions are scaled to 512×512 through interpolation and weighted averaged to generate the final semantic localization heatmap.
>
> ---
>
>
> [27] PartCLIPSeg: Understanding Multi-Granularityfor Open-Vocabulary Part Segmentation (NeurIPS 2024)
>
> [38] Image-to-Image Matching via Foundation Models: A New Perspective for Open-Vocabulary Semantic Segmentation (CVPR 2024)
>
> [45] Diffumask: Synthesizing391 images with pixel-level annotations for semantic segmentation using diffusion models (ICCV 2023)
>
> [46] OVDiff: Diffusion Models for Open-Vocabulary Segmentation (ECCV 2024)
>
> [53] PartCATSeg: Fine-Grained Image-Text Correspondence withCost Aggregation for Open-Vocabulary Part Segmentation (CVPR 2025)

---

> > ### Comment · Reviewer_76j9 · 2025-08-04
> > **Following questions.**
> >
> > Thank you for your throughout rebuttal.
> >
> > After reading your rebuttal, I have an additional question. In the W4 ablation studies, the performance improvements observed when adding HPCGraph or BAR individually to the baseline are greater than when both methods are applied together.
> > For example, on the ADE20K-Part-234 dataset, adding BAR to the baseline improves mIoU by 1.55, and adding HPCGraph improves it by 3.03. However, applying both methods together results in only a 4.11 improvement.
> > This suggests that instead of synergistically boosting each other’s performance, the two methods may actually interfere with one another. This raises some concerns about the proposed methodology, and I would appreciate it if you could address this point.

---

> > > ### Author Response · Authors · 2025-08-06
> > >
> > > Dear Reviewer,
> > >
> > > Thank you again for your thoughtful and constructive feedback on our submission. We truly appreciate the time and effort you have devoted to the review process.
> > >
> > > With only a limited time budget left in the discussion period, we would like to kindly confirm whether our previous responses have fully addressed your concerns. Your insights have been invaluable, and we’ve made every effort to respond with clarity and precision. If anything remains unclear or if further clarification would be helpful, we would be more than happy to continue the discussion.
> > >
> > > We would be sincerely grateful if you could explicitly let us know whether our explanations have resolved your concerns. Thank you once again for your time and consideration.
> > >
> > > Best regards,
> > >
> > > The authors of Paper 8092

---

> > > > ### Comment · Reviewer_76j9 · 2025-08-09
> > > >
> > > > Thank you for the detailed response. Most of my concerns have been addressed. I will take this into consideration in my final decision.

---

> ### Author Response · Authors · 2025-08-04
>
> Dear Reviewers,
>
> We sincerely thank you for your time and effort in reviewing our paper and offering valuable suggestions. As the author-reviewer discussion deadline is approaching, we would like to kindly confirm whether our previous responses have effectively addressed your concerns. We submitted detailed replies to your comments a few days ago and hope they have clarified the issues you raised. If any questions remain or if further clarification is needed, please do not hesitate to let us know. We are more than happy to continue the discussion and provide any additional information you may require.
>
> Best regards,
>
> The authors of Paper ID 8092

---

> ### Author Response · Authors · 2025-08-05
>
> Thank you very much for your prompt and thoughtful feedback. Regarding the phenomenon in the W4 ablation study where "the combined gain is less than the sum of individual gains," we provide the following explanation:
>
> First, in the ablation study, "w/ HPCGraph" refers to the introduction of hierarchical segmentation with cross-hierarchy matching (L227-229), and "w/ BAR" refers to the introduction of the BAR module to optimize the boundaries of structurally connected parts (L231-236). HPCGraph provides prior constraints on the overall structure, primarily addressing global localization errors caused by structural inconsistencies, whereas BAR focuses on locally connected regions, improving boundary segmentation accuracy through uncertainty modeling and feature refinement.
>
> **Although their design objectives differ, they may correct the same segmentation errors during actual inference.** For example, in the case of the "cat leg-torso boundary": HPCGraph uses structural priors to guide the model to first locate the overall region of the cat, then divide parts such as legs and torso within it, reducing localization errors in the boundary region from a global perspective; BAR optimizes boundary precision from a local perspective by enhancing the feature discriminability of the boundary region. **When combined, some errors are already corrected in advance by HPCGraph, leading to compressed incremental space for the BAR module. Thus, the performance improvement shows diminishing marginal returns—which is not an "interference effect" between modules but a natural result of functional complementarity.** This cross-correction capability precisely confirms the synergy of the two modules: they have clear divisions of labor in general directions (structural modeling vs. boundary optimization) and complementarily cover specific error cases, ultimately achieving overall performance improvement through the progressive logic of "structural constraints + detail optimization."
>
> To further verify the complementarity of the two modules, we designed a cross-region ablation experiment to clarify the scope of action and collaborative mechanism of each module. Based on classification uncertainty (L164-170), we divided the image prediction masks into:
> - Core Region: High-confidence, structurally clear regions with concentrated target distribution, such as body trunks and heads;
> - Boundary Region: Structurally ambiguous, texture-complex regions or junctions of multiple parts, such as limb joints and component seams.
>
> Based on the above region division, we evaluated the segmentation performance of different model configurations on the ADE20K-Part-234. As shown in the table below, **HPCGraph achieves the most significant improvement in the segmentation accuracy of core regions, verifying its ability to model the overall structure; BAR shows more obvious improvements in boundary region segmentation, confirming its refinement effect on locally ambiguous regions;** the combined model achieves the best results in both regions, further confirming the spatial complementarity of the two modules rather than redundancy or mutual interference.
>
> |**w/ HPCGraph**|**w/ BAR**|**Core Region mIoU**| **Boundary Region mIoU** |
> |:-:|:-:|:-:|:-:|
> |||24.17|18.02|
> ||✓|26.04|$\underline{21.68}$|
> |✓||$\underline{27.31}$|19.74|
> |✓|✓|**28.25**|**22.81**|
>
> In summary, although the combined gain does not numerically equal the sum of the gains from the two modules, this is a reasonable phenomenon of diminishing marginal effects and does not indicate "negative interference." In fact, the complete PBAPS framework consistently achieves the optimal performance across all tested datasets (W4 Table). HPCGraph and BAR synergistically optimize at different granularities, collectively improving segmentation results through structural modeling and boundary perception, respectively.
>
> Thank you again for your valuable feedback! If you have any further questions, we are happy to provide detailed explanations.

---

> ### Author Response · Authors · 2025-08-09
>
> Thank you for your time and thoughtful feedback. We appreciate your consideration and are grateful for the opportunity to address your concerns.

---

### Official Review · Reviewer_LruU · 2025-07-09

**Clarity:** 3
**Significance:** 2
**Originality:** 3
**Rating:** 4
**Confidence:** 4

**Summary:**

This paper addresses the problem of open-vocabulary part segmentation (OVPS), focusing on the challenge of segmenting structurally connected parts that lack clear visual boundaries. The authors argue that standard discrete classification struggles with these smooth transitions.

To mitigate this, they propose a Boundary-Aware Refinement (BAR) module that identifies ambiguous boundaries and refines them using nearby reliable regions and adaptive prototypes. They further design a training-free framework called PBAPS, which performs progressive refinement guided by a hierarchical graph of part relationships.

The method achieves strong performance on several benchmarks. While the use of multiple external models may introduce complexity, the approach is well-motivated and effectively tackles a core limitation in current OVPS systems.

**Questions:**

See weakness.

**Ethical Concerns:**

["NO or VERY MINOR ethics concerns only"]

**Final Justification:**

The authors have addressed my concern regarding the relationship between the prototype and the test set data and results. I am willing to raise my score accordingly.

**Limitations:**

Yes

**Paper Formatting Concerns:**

No.

**Quality:**

3

**Strengths And Weaknesses:**

Strengths:

- The method is entirely training-free, avoiding the need for large-scale annotated data or fine-tuning.

- The Boundary-Aware Refinement (BAR) module is a novel design that specifically targets the challenge of ambiguous part boundaries, a key limitation in existing OVPS methods.

- The use of a hierarchical part graph to guide progressive refinement introduces a structured and interpretable approach to part segmentation.

- The framework is modular and leverages existing vision-language models effectively without additional training overhead.

Weaknesses:

- The method relies on generating hundreds of prototype images per part class, which may introduce a risk of overfitting or inadvertently aligning too closely with the test set, especially on relatively small or simple datasets. It would be helpful to demonstrate the method on more complex or diverse scenarios

- The paper lacks an analysis of how the similarity between generated prototypes and dataset images affects performance, which would be important to clarify whether the method generalizes or exploits dataset biases.

---

> ### Author Rebuttal · Authors · 2025-07-30
>
> We thank reviewer for the constructive comments. We provide our feedbacks as follows.
>
> ---
> >**W1: The method relies on generating hundreds of prototype images per part class, which may introduce a risk of overfitting or inadvertently aligning too closely with the test set, especially on relatively small or simple datasets. It would be helpful to demonstrate the method on more complex or diverse scenarios.**
>
> **A1:** Thank you for pointing out this important concern regarding the potential proximity between generated prototypes and the test set. We provide the following clarifications:
> - The prototype images are entirely synthesized by Stable Diffusion without using any images from the target datasets. All part prototype images are generated using the publicly available pre-trained text-to-image model (Stable Diffusion v1.4). **These synthesized prototypes differ significantly from real-world test images in style, background, and composition**, thereby minimizing the risk of overfitting or unintended alignment with the test set.
> - To quantitatively assess the distinctness between the synthesized prototypes and the test set images, we conducted a **feature-level similarity analysis**. First, we input the prototype set and test set images into DINOv2 ViT-B/16 to extract global image features. Then, we performed L2 normalization on these image features and calculated the mean values of the features of the prototype set and test set respectively as their central vectors. The cosine similarity between these two central vectors was used to quantify the similarity between the prototype set and the test set. For comparison, we also calculated the similarity within the test set. We randomly split the test set into two halves: one half was used to generate a central vector $\mathbf{V}_1$, whose similarity with the prototype set’s central vector $\mathbf{V}$ was calculated. The other half was used to generate a central vector $\mathbf{V}_2$, whose similarity with $\mathbf{V}_1$ was calculated as a reference for the consistency of the test set. The results are shown in the table below. The similarity between the prototype set and the test set is significantly lower than the internal similarity of the test set, indicating that the synthesized prototypes are significantly different from the test set images and there is no risk of "excessive proximity".
> | Similarity Type | **Pascal-Part-116** | **ADE20K-Part-234** | **PartImageNet**|
> |:-|:-:|:-:|:-:|
> | Prototype set vs. Test set |0.63 ± 0.01|0.59 ± 0.01|0.62 ± 0.01|
> | Within Test set |0.87 ± 0.01|0.86 ± 0.02|0.89 ± 0.01|
>
> - In terms of generalizability, PBAPS has been comprehensively evaluated on three highly diverse datasets: Pascal-Part-116 (natural objects such as animals), ADE20K-Part-234 (indoor scenes with furniture and appliances), and PartImageNet (cross-domain categories like vehicles and instruments). These datasets vary significantly in part types, scene complexity, and domain coverage. As shown in Table 2, using a single prototype generation process and fixed hyperparameters, **PBAPS achieves state-of-the-art results across all three datasets**, validating its strong cross-domain generalization ability.
>
> In summary, the prototype generation process in PBAPS is entirely independent of the target datasets and does not introduce any data leakage risk. Our quantitative analysis confirms that the synthesized prototypes are significantly distinct from the test images, alleviating concerns of overfitting or excessive similarity. Furthermore, the consistent performance across diverse datasets demonstrates the robustness and generalizability of our method, highlighting its suitability for large-scale and cross-domain open-vocabulary part segmentation tasks.
>
> >**W2: The paper lacks an analysis of how the similarity between generated prototypes and dataset images affects performance, which would be important to clarify whether the method generalizes or exploits dataset biases.**
>
> **A2:** Thank you for highlighting the potential impact of prototype-test image similarity on model generalization. We provide the following clarifications and additional analysis:
> - It should be emphasized that the approach of constructing visual prototypes using Stable Diffusion + SAM + DINO has been adopted and validated as effective in multiple existing works (e.g., OVDiff [46], RIM [38]). **Our contribution does not lie in the prototype construction, but rather in addressing the challenge of low segmentation accuracy at structurally connected part boundaries in OVPS.** Guided by HPCGraph, hierarchical segmentation is performed, and the BAR module improves boundary precision via feature optimization and dynamic prototype mechanisms. This design is the key reason why our method PBAPS outperforms the state-of-the-art method RIM by **7.3%, 18.1%**, and **8.4%** on Pascal-Part-116, ADE20K-Part-234, and PartImageNet, respectively (as shown in Table 2).
> - Although the prototype generation mechanism is not the contribution of this paper, we agree that this mechanism may affect model performance. Following your suggestion, we designed experiments to **analyze the relationship between "prototype image-test set similarity" and model performance (mIoU).** First, we synthesized prototype images with "low similarity" (<0.3) to real test set images by modifying the style prompts of Stable Diffusion (e.g., "cartoon"). Then, we slightly augmented and perturbed the training set images of the target datasets to generate prototype images with "high similarity" (>0.8). We evaluated the performance of PBAPS using prototype features generated from these two types of prototype images. The results are shown in the table below. Segmentation performance is positively correlated with prototype similarity, which is reasonable and expected. This is because prototypes with high similarity can more easily provide discriminative visual features, enabling more accurate segmentation. However, using prototypes with high similarity to the target dataset may weaken the model's cross-domain generalizability and make it less robust to style or distribution shifts.
> | Prototype-Test Similarity | **Pascal-Part-116** | **ADE20K-Part-234** | **PartImageNet**|
> |:-|:-:|:-:|:-:|
> | Low (<0.3)|43.20|21.87|38.62|
> | Our method (0.55~0.65) |46.35|24.70|42.61|
> | High (>0.8)| 50.83|29.04|47.06|
> - To address the discrepancies between the prototype set and the test set in terms of style, distribution, or visual details, relying solely on static visual prototypes may lead to inaccurate matching or misaligned responses, ultimately affecting the precision of part localization. To mitigate this issue, the BAR module introduces a **Dynamic Prototype Adaptation mechanism, which effectively enhances the adaptability of prototypes to test images** and improves segmentation performance. As described in Lines 182–189, this mechanism fuses the global prototype with high-confidence region features extracted from the current image to generate an adaptive prototype that better aligns with the actual content of the image. Compared to using only pre-generated static prototypes, this fusion strategy enables the model to capture instance-specific variations and contextual cues, making the prototype more consistent with the spatial distribution and structural characteristics of the target part. As shown in the ablation study (Table 3), removing the dynamic prototype adaptation mechanism leads to a significant drop in mIoU and bIoU across all three datasets, further confirming its effectiveness in improving segmentation accuracy and mitigating the impact of cross-image discrepancies.
>
> In summary, while prototype similarity does influence segmentation performance, our method is carefully designed to **maintain a balance between effectiveness and generalizability.** The use of synthesized prototypes with moderate similarity, combined with the BAR module's dynamic prototype adaptation mechanism, ensures robust performance without overfitting to specific datasets. These design choices allow PBAPS to generalize well across diverse domains while preserving high segmentation accuracy.
>
> ---
> We once again sincerely thank the reviewer for the insightful and constructive feedback. If there are any further concerns or points requiring clarification, we would be glad to provide additional details.
>
> [38] Image-to-Image Matching via Foundation Models: A New Perspective for Open-Vocabulary Semantic Segmentation (CVPR 2024)
>
> [46] OVDiff: Diffusion Models for Open-Vocabulary Segmentation (ECCV 2024)

---

> > ### Author Response · Authors · 2025-08-06
> >
> > Dear Reviewer,
> >
> > Thank you again for your thoughtful and constructive feedback on our submission. We truly appreciate the time and effort you have devoted to the review process.
> >
> > With only a limited time budget left in the discussion period, we would like to kindly confirm whether our previous responses have fully addressed your concerns. Your insights have been invaluable, and we’ve made every effort to respond with clarity and precision. If anything remains unclear or if further clarification would be helpful, we would be more than happy to continue the discussion.
> >
> > We would be sincerely grateful if you could explicitly let us know whether our explanations have resolved your concerns. Thank you once again for your time and consideration.
> >
> > Best regards,
> >
> > The authors of Paper 8092

---

> ### Author Response · Authors · 2025-08-04
>
> Dear Reviewers,
>
> We sincerely thank you for your time and effort in reviewing our paper and offering valuable suggestions. As the author-reviewer discussion deadline is approaching, we would like to kindly confirm whether our previous responses have effectively addressed your concerns. We submitted detailed replies to your comments a few days ago and hope they have clarified the issues you raised. If any questions remain or if further clarification is needed, please do not hesitate to let us know. We are more than happy to continue the discussion and provide any additional information you may require.
>
> Best regards,
>
> The authors of Paper ID 8092

---

### Official Review · Reviewer_BZzg · 2025-07-14

**Clarity:** 3
**Significance:** 3
**Originality:** 3
**Rating:** 4
**Confidence:** 4

**Summary:**

This paper presents PBAPS, a training-free framework for open-vocabulary part segmentation (OVPS) that addresses the inherent challenge of segmenting structurally connected part boundaries-regions where anatomically adjacent parts exhibit smooth, continuous visual transitions. The authors identify that the core difficulty arises from the mismatch between continuous image features and discrete classification mechanisms, leading to ambiguous boundaries. To resolve this issue, PBAPS introduces a progressive boundary-aware strategy built upon Cross-Model Prototype Induction, Hierarchical Part-Connected Graph and Boundary-Aware Refinement. Extensive experiments on Pascal-Part-116, ADE20K-Part-234, and PartImageNet demonstrate state-of-the-art performance.

**Questions:**

1. Contamination from blended-part features in the “deterministic” pixels of BAR
BAR refines ambiguous boundaries by aggregating context from “deterministic” regions, yet gradual transitions may still introduce mixed-part features into these pixels. Could the authors provide  qualitative visualizations  that demonstrate this blending does **not** materially degrade final segmentation accuracy?
2. Dynamic extensibility of a static HPCGraph
Hand-crafted hierarchies (e.g., cat → head → eye) struggle with irregular or novel topologies (e.g., modular furniture). Have the authors considered a lightweight mechanism that allows the hierarchy to adapt at inference time while preserving the training-free constraint?
3. Additional supervised open-vocabulary part-detection baseline
Recent supervised open-vocabulary detectors can be minimally adapted for part-level segmentation. Please report the  mIoU/bIoU after fine-tuning at least one such model on the same part vocabulary; if PBAPS still significantly outperforms, this will strengthen its training-free advantage.
4. Ablation decoupling hierarchical reasoning from boundary refinement
The current ablation (Table 3) does not disentangle the contributions of the hierarchy and BAR. Please add a  flat baseline that applies BAR directly to DINOv2 pixel features  without HPCGraph.

**Ethical Concerns:**

["NO or VERY MINOR ethics concerns only"]

**Final Justification:**

The rebuttal has well addressed my concerns. I am glad to arise my score.

**Limitations:**

Please see weaknesses and questions.

**Quality:**

3

**Strengths And Weaknesses:**

Strengths
Quality: The paper presents a technically sound framework (PBAPS) for open-vocabulary part segmentation (OVPS). The methodology is well-structured, combining hierarchical reasoning (HPCGraph) with a novel Boundary-Aware Refinement (BAR) module. The experiments are comprehensive, covering three benchmarks (Pascal-Part-116, ADE20K-Part-234, PartImageNet) and demonstrating consistent outperformance over prior methods (e.g., +10% mIoU over PartCLIPSeg). The ablation studies validate the necessity of each component
Clarity: The paper is well-organized, with clear explanations of the BAR module’s four-step process (ambiguous region localization, feature optimization, prototype adaptation, reclassification). Figures effectively illustrate the boundary refinement pipeline. The distinction between structurally vs. non-structurally connected boundaries is rigorously analyzed via feature gradients, providing theoretical grounding.

Significance: The work addresses a critical gap in OVPS—structurally connected part boundaries—which prior methods struggle with due to smooth feature transitions.
Originality: While cross-model prototype induction resembles RIM’s pipeline, the novelty lies in adapting it for part-level segmentation and introducing BAR to resolve boundary ambiguities via cost-divergence maps and context-driven refinement. The HPCGraph’s hierarchical design (object → part → subpart) is tailored to OVPS, distinguishing it from prior OVSS methods.


Weaknesses
Over-smoothing in BAR
BAR averages context from “certain” pixels to refine boundaries, but these pixels may themselves be ambiguous on gradual transitions, weakening discriminative cues.
 HPCGraph static dependency
The hand-crafted hierarchy (e.g., cat → head → eye) cannot generalize to objects with irregular or novel part topologies (e.g., modular furniture).

Missing comparison to supervised open-vocabulary part detectors
The experimental baseline set omits recent strong supervised open-vocabulary detectors that can be adapted for part localization. Without this comparison, the paper cannot fully substantiate the added value of its training-free approach over competitive fine-tuned alternatives.

Inadequate ablation separating hierarchical reasoning and boundary refinement
The ablation study (Table 3) entangles the contributions of (i) the hierarchical HPCGraph and (ii) the BAR boundary-refinement module. A crucial baseline—BAR applied directly to flat DINOv2 features without the hierarchy—is absent, preventing an isolated assessment of each component’s impact.

---

> ### Author Rebuttal · Authors · 2025-07-30
>
> We thank reviewer for the constructive comments. We provide our feedbacks as follows.
>
> ---
> >**W1＆Q1: Contamination from blended-part features in the “deterministic” pixels of BAR.**
>
> **A1:** Regarding the question of whether BAR weakens the original discriminative features, we provide the following supplementary explanation:
> - In terms of design, BAR does not directly perform average context fusion on ambiguous regions. The attention weights in Equation 4 **only aggregate features from deterministic regions** that are most relevant to the current ambiguous pixel, avoiding the introduction of irrelevant or ambiguous contexts. Subsequently, a weighted fusion strategy of "original features ($\gamma$=0.8) + context (1-$\gamma$=0.2)" is adopted as specified in Equation 5, ensuring that **the discriminative features of the pixels remain dominant**. This design not only leverages context to resolve ambiguity but also prevents the loss of discriminative features.
> - In Appendix B.4, we evaluated the model performance under different $\gamma$ values. As shown in Figure 8(a), using only the original ambiguous region features (without context fusion, $\gamma$=1.0) results in 44.83% mIoU. Using only context features (without original features, $\gamma$=0),  the mIoU drops to 43.31% (not included in the figure), indicating the loss of discriminative features and excessive boundary smoothing. With balanced fusion ($\gamma$=0.8), the mIoU increases to 46.35%, which **retains discriminative features while eliminating ambiguity through context.**
> - Additionally, the ablation experiment in Table 3 effectively verifies the contribution of the BAR module. Figure 4 provides a visual comparison of the segmentation at the "cat neck-torso boundary" before and after refinement. Due to NeurIPS rebuttal constraints, we are unable to include images, PDFs, or external links at this stage. We will supplement additional qualitative visualization results in subsequent revisions of the paper to fully demonstrate the effectiveness of the BAR module.
>
> >**W2＆Q2: Extensibility of the HPCGraph.**
>
> **A2:** Regarding the Extensibility of HPCGraph, we provide the following supplementary explanations:
> - HPCGraph is constructed based on commonsense structural knowledge and does not rely on any manual annotations. As such, it incurs low construction cost and exhibits strong transferability and scalability. In large-scale object scenarios, **both the topological structure and visual prototypes can be shared and reused.** For novel object categories with similar structures to existing ones (e.g., “zebra” as a novel category relative to the existing category “horse”), we can directly reuse existing topologies (e.g., “zebra head ↔ zebra neck” correspond to the structure of "horse"). For PartImageNet (L198–200), we divided 158 object categories into 11 superclasses, where each superclass shares a unified topological graph. This strategy reduces the number of topology by **93%** (from 158 to 11) while maintaining high performance—PBAPS outperforms RIM by **8.4%** on PartImageNet (Table 2). When the number of object categories expands to tens of thousands, a large number of them are highly similar in form and structure (such as horses, donkeys, mules, zebras, etc.). We can use LLMs to analyze inter-category structural similarities and achieve cluster-level topology reuse (i.e., categories with similar structures share a set of topologies). Furthermore, part visual prototypes can also be reused (e.g., the prototype for “horse leg” can be applied to “donkey leg”).
> - In contrast to existing methods such as PartCLIPSeg [27] and PartCATSeg [53], which require extensive part-level annotations and retraining when introducing new object categories, our method is training-free and supervision-independent. It supports open-world scalability by simply extending the HPCGraph and adding a small number of visual prototypes.
> - As noted in Appendix A.1, when dealing with objects with irregular structures (e.g., modular furniture), the static graph structure has insufficient adaptability. To address this, a dynamic graph generation mechanism can be considered. The core logic is to **automatically add adjacency edges for highly relevant parts based on the spatial adjacency and visual feature similarity between part response regions**. The following table compares performance using static and dynamic graphs. While static graphs yield better performance in well-structured domains, the dynamic version provides competitive results and significantly enhances flexibility for deployment in unconstrained real-world scenarios.
> | HPCGraph | **Pascal-Part-116**  | **ADE20K-Part-234** | **PartImageNet**|
> |:-|:-:|:-:|:-:|
> |  | mIoU&ensp;&emsp;bIoU | mIoU&ensp;&emsp;bIoU| mIoU&ensp;&emsp;bIoU |
> | Dynamic | 44.63&ensp;&emsp;33.12 |23.87&ensp;&emsp;15.63| 41.66&ensp;&emsp;28.72 |
> | Static | **46.35**&ensp;&emsp;**34.46** |**24.70**&ensp;&emsp;**16.41** | **42.61**&ensp;&emsp;**29.31** |
>
>
> >**W3＆Q3: Additional supervised open-vocabulary part-detection baseline.**
>
> **A3:** Regarding the comparison with supervised open-vocabulary segmentation methods, as shown in Table 2, our method (PBAPS) has already compared with ZSSeg (ECCV 2022) [15] and CAT-Seg (CVPR 2024) [19], and outperformed these methods on all three datasets, effectively verifying the superiority of our training-free approach. Following your suggestion, we have further compared with two latest supervised open-vocabulary segmentation methods, SED (CVPR 2024) [60] and DPSeg (CVPR 2025) [61]. Both methods introduce a multi-scale feature fusion strategy based on CAT-Seg, achieving significant improvements in the segmentation of small objects and boundaries. We adjusted their decoders to support part-level prediction and fine-tuned them on the target datasets. As shown in the table below, **PBAPS still maintains superior performance**, further validating its advantages in OVPS.
> | Method | **Pascal-Part-116**  | **ADE20K-Part-234** | **PartImageNet**|
> |:-|:-:|:-:|:-:|
> |  | mIoU&ensp;&emsp;bIoU | mIoU&ensp;&emsp;bIoU| mIoU&ensp;&emsp;bIoU |
> | SED | 32.61&ensp;&emsp;21.76 |12.74&emsp;&emsp;8.58| 30.20&ensp;&emsp;19.45 |
> | DPSeg | 34.64&ensp;&emsp;22.49 |13.21&emsp;&emsp;8.87 | 32.57&ensp;&emsp;21.97 |
> | PBAPS (ours) | **46.35**&ensp;&emsp;**34.46** |**24.70**&ensp;&emsp;**16.41** | **42.61**&ensp;&emsp;**29.31** |
>
> >**W4＆Q4: Ablation decoupling hierarchical reasoning from boundary refinement.**
>
> **A4:** As shown in the table below, we directly applied BAR to optimize all structurally connected part boundaries based on the baseline (the 1st row, which excludes all components and directly matches pixel-wise features with visual prototypes). Furthermore, to further demonstrate the robustness and generalizability of our method, we supplement the same ablation experiment on ADE20K-Part-234 and PartImageNet. The results indicate that without introducing hierarchical structural constraints, BAR can still effectively model local contextual consistency and improve the segmentation accuracy of boundaries between structurally connected parts. Moreover, combining BAR with hierarchical reasoning yields further improvements, demonstrating their complementarity.
> | w/ HPCGraph |w/ BAR| **Pascal-Part-116**  | **ADE20K-Part-234** | **PartImageNet**|
> |:-:|:-:|:-:|:-:|:-:|
> | | | mIoU&ensp;&emsp;bIoU | mIoU&ensp;&emsp;bIoU| mIoU&ensp;&emsp;bIoU |
> |  |  |40.74&ensp;&emsp;31.09 |20.59&ensp;&emsp;13.72 | 37.46&ensp;&emsp;25.78 |
> |  | √ |42.95&ensp;&emsp;32.34 |22.14&ensp;&emsp;14.53 | 39.54&ensp;&emsp;27.12 |
> | √ |  |44.08&ensp;&emsp;32.99 |23.62&ensp;&emsp;15.43 | 41.27&ensp;&emsp;28.58 |
> | √ | √ |46.35&ensp;&emsp;34.46 |24.70&ensp;&emsp;16.41 | 42.61&ensp;&emsp;29.31 |
>
> ---
> We once again sincerely thank the reviewer for the insightful and constructive feedback. If there are any further concerns or points requiring clarification, we would be glad to provide additional details.
>
> [15] ZSSeg: A Simple Baseline for Open-Vocabulary Semantic Segmentation with Pre-trained Vision-language Model (ECCV 2022)
>
> [19] CAT-Seg: Cost Aggregation for Open-Vocabulary Semantic Segmentation (CVPR 2024)
>
> [60] SED: ASimple Encoder-Decoder for Open-Vocabulary Semantic Segmentation (CVPR2024)
>
> [61] DPSeg: Dual-Prompt Cost Volume Learning for Open-Vocabulary Semantic Segmentation (CVPR2025)

---

> > ### Author Response · Authors · 2025-08-06
> >
> > Dear Reviewer,
> >
> > Thank you again for your thoughtful and constructive feedback on our submission. We truly appreciate the time and effort you have devoted to the review process.
> >
> > With only a limited time budget left in the discussion period, we would like to kindly confirm whether our previous responses have fully addressed your concerns. Your insights have been invaluable, and we’ve made every effort to respond with clarity and precision. If anything remains unclear or if further clarification would be helpful, we would be more than happy to continue the discussion.
> >
> > We would be sincerely grateful if you could explicitly let us know whether our explanations have resolved your concerns. Thank you once again for your time and consideration.
> >
> > Best regards,
> >
> > The authors of Paper 8092

---

> ### Author Response · Authors · 2025-08-04
>
> Dear Reviewers,
>
> We sincerely thank you for your time and effort in reviewing our paper and offering valuable suggestions. As the author-reviewer discussion deadline is approaching, we would like to kindly confirm whether our previous responses have effectively addressed your concerns. We submitted detailed replies to your comments a few days ago and hope they have clarified the issues you raised. If any questions remain or if further clarification is needed, please do not hesitate to let us know. We are more than happy to continue the discussion and provide any additional information you may require.
>
> Best regards,
>
> The authors of Paper ID 8092

---

### Official Review · Reviewer_Sy1A · 2025-07-21

**Clarity:** 3
**Significance:** 3
**Originality:** 3
**Rating:** 4
**Confidence:** 3

**Summary:**

This paper proposes a novel training-free Open-Vocabulary Part Segmentation (OVPS) framework called PBAPS, addressing the inaccuracies in segmenting structurally connected part boundaries caused by the conflict between continuous image features and discrete classification mechanisms. PBAPS employs a progressive segmentation strategy guided by a Hierarchical Part Connected Graph (HPCGraph) and introduces a Boundary-Aware Refinement (BAR) module. The BAR module locates ambiguous boundary regions, optimizes pixel features, and adaptively refines visual prototypes for precise reclassification. Experimental results demonstrate that PBAPS significantly outperforms state-of-the-art methods on multiple benchmark datasets, showcasing its superior performance, especially in handling fine-grained part boundaries.

**Questions:**

see weakness above

**Ethical Concerns:**

["NO or VERY MINOR ethics concerns only"]

**Limitations:**

see weakness above

**Paper Formatting Concerns:**

No formatting concerns.

**Quality:**

3

**Strengths And Weaknesses:**

Strengths：
1. This paper proposes a framework that successfully leverages pre-trained foundation models for a complex downstream task without any fine-tuning, demonstrating excellent zero-shot generalization and highlighting a promising research direction.
2. The method achieves impressive SOTA results on three challenging benchmarks.
Weakness:
1. The paper only showcases successful examples. A discussion of failure cases would provide a more balanced perspective on the method's limitations and offer crucial insights for future work.
2. The paper lacks a clear justification for how key hyperparameters were chosen without access to the test set. Furthermore, their robustness and generalizability across different datasets are not discussed.
3. The method's reliance on a manually constructed Hierarchical Part Connected Graph (HPCGraph) is a major bottleneck, raising significant concerns about its scalability and applicability to novel object categories not defined in advance.

---

> ### Author Rebuttal · Authors · 2025-07-30
>
> We thank reviewer for the constructive comments. We provide our feedbacks as follows.
>
> ---
> >**W1: The paper only showcases successful examples. A discussion of failure cases would provide a more balanced perspective on the method's limitations and offer crucial insights for future work.**
>
> **A1:** We fully agree with your comment. In the main paper, we primarily showcased representative successful cases to clearly demonstrate the advantages of our method. Although appendix A.1 includes a discussion on the limitations of our approach—such as difficulties in distinguishing multiple instances and handling complex or irregular object structures—it lacks visual examples that intuitively illustrate failure cases. Due to NeurIPS rebuttal constraints, we are unable to include images, PDFs, or external links at this stage. We briefly describe the failure cases here: when there are multiple animals of the same category in an image (e.g., three cats) with intertwined limbs, PBAPS fails to distinguish "identical parts belonging to different individuals" (e.g., the tail of a black cat vs. the tail of a white cat), resulting in cross-instance confusion. For modular furniture or mechanical parts, whose part hierarchies are ambiguous or topologies are variable, the static HPCGraph struggles to cover such scenarios, leading to part missed detections or over-merging (e.g., misclassifying folded table legs as part of the tabletop). We will include visualizations of typical failure cases in the revised version of the paper and provide an in-depth analysis of the underlying causes.
>
> >**W2: The paper lacks a clear justification for how key hyperparameters were chosen without access to the test set. Furthermore, their robustness and generalizability across different datasets are not discussed.**
>
> **A2:** Thank you for your valuable comment. Regarding hyperparameter selection, we strictly adhered to the **"no access to test set"** principle. Specifically, we evaluated model performance (mIoU) under different hyperparameter configurations on the training sets of the three target datasets. As shown in the table below, we selected a single set of hyperparameters ($\gamma$ = 0.8, $\alpha$ = 0.7, $K$ = 4) that consistently performed well across all three datasets. **This selection was made without any feedback from the test sets, thus avoiding dataset-specific tuning.** As for robustness, Table 2 demonstrates that PBAPS (ours), using the same hyperparameter configuration, achieves state-of-the-art performance on Pascal-Part-116, ADE20K-Part-234, and PartImageNet—datasets with significant differences in visual style and structural granularity—validating the robustness and generalizability of our hyperparameter choices across diverse domains.
> | &emsp;$\gamma$&emsp; | **Pascal-Part-116** | **ADE20K-Part-234** | **PartImageNet**|
> |:-:|:-:|:-:|:-:|
> | 0.5 |43.82|23.76|40.21|
> | 0.6 |44.73|24.31|40.84|
> | 0.7 |45.51|**24.79**|42.39|
> | 0.8 |**46.21**|24.62|**42.75**|
> | 0.9 |45.40|24.03|41.42|
> | 1.0 |44.72|23.89|40.94|
>
> |&emsp; $\alpha$&emsp; | **Pascal-Part-116** | **ADE20K-Part-234** | **PartImageNet**|
> |:-:|:-:|:-:|:-:|
> | 0.5 |45.67|24.08|41.62|
> | 0.6 |46.02|24.39|42.16|
> | 0.7 |**46.21**|**24.62**|**42.75**|
> | 0.8 |46.12|24.46|42.08|
> | 0.9 |45.83|24.25|41.86|
> | 1.0 |45.64|24.06|41.51|
>
> |&emsp; $K$&emsp; | **Pascal-Part-116** | **ADE20K-Part-234** | **PartImageNet**|
> |:-:|:-:|:-:|:-:|
> | 1 |45.21|24.15|41.43|
> | 2 |45.91|24.43|41.87|
> | 4 |**46.21**|**24.62**|**42.75**|
> | 6 |45.88|24.48|41.91|
> | 8 |45.62|24.22|41.54|
>
> >**W3: The method's reliance on a manually constructed Hierarchical Part Connected Graph (HPCGraph) is a major bottleneck, raising significant concerns about its scalability and applicability to novel object categories not defined in advance.**
>
> **A3:** Regarding the scalability of HPCGraph, we provide the following supplementary explanations:
> - HPCGraph is constructed based on commonsense structural knowledge (e.g., compositional relationships, anatomical structures) and does not rely on any manual annotations. As such, it incurs low construction cost and exhibits strong transferability and scalability.
> - **In large-scale object scenarios, both the topological structure and visual prototypes can be shared and reused.** For novel object categories with similar structures to existing ones (e.g., “zebra” as a novel category relative to the existing category “horse”), we can directly reuse existing topologies (e.g., “zebra head → zebra eye”, “zebra head ↔ zebra neck” correspond to the structure of "horse"). For PartImageNet (L198–200), we divided 158 object categories into 11 superclasses, where each superclass shares a unified topological graph. This strategy reduces the number of topology by **93%** (from 158 to 11) while maintaining high performance—PBAPS outperforms RIM by **8.4%** on PartImageNet (Table 2). When the number of object categories expands to tens of thousands, a large number of them are highly similar in form and structure (such as horses, donkeys, mules, zebras, etc.). We can use Large Language Models (LLMs) to analyze inter-category structural similarities and achieve cluster-level topology reuse (i.e., categories with similar structures share a set of topologies). Furthermore, part visual prototypes can also be reused (e.g., the prototype for “horse leg” can be applied to “donkey leg”).
> - In contrast to existing methods such as PartCLIPSeg [27] and PartCATSeg [53], which require extensive part-level annotations and retraining when introducing new object categories, our method is training-free and supervision-independent. It supports open-world scalability by simply extending the HPCGraph and adding a small number of visual prototypes, making it more suitable for large-scale and dynamic settings.
> - As noted in Appendix A.1, when dealing with objects with irregular structures (e.g., modular furniture), the static graph structure has insufficient adaptability. To address this, a dynamic graph generation mechanism can be considered. The core logic is to **automatically add adjacency edges for highly relevant parts based on the spatial adjacency and visual feature similarity between part response regions**. The following table compares performance using static and dynamic graphs. While static graphs yield better performance in well-structured domains, the dynamic version provides competitive results and significantly enhances flexibility for deployment in unconstrained real-world scenarios.
> | HPCGraph | **Pascal-Part-116**  | **ADE20K-Part-234** | **PartImageNet**|
> |:-|:-:|:-:|:-:|
> |  | mIoU&ensp;&emsp;bIoU | mIoU&ensp;&emsp;bIoU| mIoU&ensp;&emsp;bIoU |
> | Dynamic | 44.63&ensp;&emsp;33.12 |23.87&ensp;&emsp;15.63| 41.66&ensp;&emsp;28.72 |
> | Static | **46.35**&ensp;&emsp;**34.46** |**24.70**&ensp;&emsp;**16.41** | **42.61**&ensp;&emsp;**29.31** |
>
> ---
> We once again sincerely thank the reviewer for the insightful and constructive feedback. If there are any further concerns or points requiring clarification, we would be glad to provide additional details.
>
> [27] PartCLIPSeg: Understanding Multi-Granularityfor Open-Vocabulary Part Segmentation (NeurIPS 2024)
>
> [53] PartCATSeg: Fine-Grained Image-Text Correspondence withCost Aggregation for Open-Vocabulary Part Segmentation (CVPR 2025)

---

> > ### Author Response · Authors · 2025-08-06
> >
> > Dear Reviewer,
> >
> > Thank you again for your thoughtful and constructive feedback on our submission. We truly appreciate the time and effort you have devoted to the review process.
> >
> > With only a limited time budget left in the discussion period, we would like to kindly confirm whether our previous responses have fully addressed your concerns. Your insights have been invaluable, and we’ve made every effort to respond with clarity and precision. If anything remains unclear or if further clarification would be helpful, we would be more than happy to continue the discussion.
> >
> > We would be sincerely grateful if you could explicitly let us know whether our explanations have resolved your concerns. Thank you once again for your time and consideration.
> >
> > Best regards,
> >
> > The authors of Paper 8092

---

> ### Author Response · Authors · 2025-08-04
>
> Dear Reviewers,
>
> We sincerely thank you for your time and effort in reviewing our paper and offering valuable suggestions. As the author-reviewer discussion deadline is approaching, we would like to kindly confirm whether our previous responses have effectively addressed your concerns. We submitted detailed replies to your comments a few days ago and hope they have clarified the issues you raised. If any questions remain or if further clarification is needed, please do not hesitate to let us know. We are more than happy to continue the discussion and provide any additional information you may require.
>
> Best regards,
>
> The authors of Paper ID 8092

---

### Author Response · Authors · 2025-08-07

Dear Reviewers,

As the discussion phase draws to a close, we’d like to kindly follow up to ensure that our previous responses have reached you.

We sincerely appreciate the time and care you’ve devoted to reviewing our submission during this busy period. If you’ve had a chance to review our earlier replies, we would be grateful to know whether they have adequately addressed your concerns.

If any part of our response remains unclear or would benefit from further clarification, please don’t hesitate to let us know — we would be more than happy to provide additional details. Your insights are invaluable to us, and we remain fully committed to addressing any remaining questions.

Thank you once again for your time and thoughtful consideration.

Best regards,

The authors of Paper 8092

---

### Note · Authors · 2025-08-14

Dear AC and Reviewers,

We offer our final remarks on the review process. First, **we sincerely thank Reviewer 76j9 and 4fTr for their active engagement**—their insightful questions and suggestions helped refine our work by clarifying contributions, addressing limitations, and prompting additional experiments and mechanism explanations. **Though we unfortunately did not have the chance to discuss with Reviewer Sy1A, BZzg, and LruU, we still express our sincere gratitude to them and emphasize that we have provided comprehensive responses to their questions** :

- Reviewer Sy1A: For the "discussion of failure cases", we described typical multi-instance confusion and irregular object errors, analyzed causes, and committed to adding visualizations. For "hyperparameter selection," we explained parameters were chosen solely from training sets of three datasets, achieving SOTA across diverse datasets, showing robustness. For "HPCGraph scalability", we validated it via topology reuse, prototype reuse, and a dynamic graph mechanism.

- Reviewer BZzg: On "BAR over-smoothing", we clarified BAR aggregates only the most relevant high-confidence features and uses weighted fusion to retain discriminative features while reducing ambiguity. On "dynamic scalability", we introduced a dynamic graph mechanism that performs comparably to the static graph in unstructured scenarios. On "comparison with supervised methods", we added SED (CVPR 2024) and DPSeg (CVPR 2025) results, where PBAPS leads in mIoU and bIoU. Additional ablations showed HPCGraph and BAR are complementary, with best performance achieved when combined.

- Reviewer LruU: On "overfitting from prototype generation", similarity analysis showed prototype–test set similarity is far lower than intra-test set similarity, and prototype images generated by Stable Diffusion are significantly different from the test set, avoiding overfitting. On "similarity–performance relation", experiments show medium-similarity prototypes with BAR's prototype adaptation mechanism balance accuracy and generalization, performing stably on cross-domain datasets.

We believe these responses have fully addressed the questions raised by the reviewers. Any remaining unclear points will be clarified in the revision. We again thank all reviewers for their time and effort, especially 76j9 and 4fTr for their constructive engagement. We look forward to the AC's final evaluation of this research.

Best regards,

The authors of Paper ID 8092

---

### Decision · Program_Chairs · 2025-09-17

**Decision:**

Accept (poster)

**Comment:**

This paper introduces PBAPS, a novel and effective training-free framework for open-vocabulary part segmentation that achieves state-of-the-art results on several benchmarks. The core contribution includes identifying and addressing the inaccurate segmentation of structurally connected part boundaries, which is a key weakness in prior methods. The authors propose a progressive, hierarchically-guided segmentation strategy using a Hierarchical Part Connected Graph (HPCGraph) and a Boundary-Aware Refinement (BAR) module, which intelligently identifies and refines ambiguous boundary regions. The primary strengths of the paper are its novel training-free approach, the technical soundness of the proposed modules, and the significant performance improvements demonstrated through comprehensive experiments. The main weaknesses initially identified by reviewers (e.g., Sy1A, BZzg, 4fTr) centered on the scalability of the manually constructed HPCGraph, the clarity of the writing, and proper attribution to prior work for prototype generation. However, the authors provided a thorough rebuttal that convincingly addressed these concerns. They demonstrated the feasibility of scaling the HPCGraph through topology and prototype reuse, supported by experiments, and proposed leveraging knowledge bases like WordNet. They also committed to significant revisions to improve readability and clarify the relationship with existing methods, even providing a revised abstract to reviewer 4fTr. Furthermore, they supplied additional ablation studies and comparisons to the latest supervised methods as requested by reviewers BZzg and 76j9, leading reviewers BZzg and LruU to raise their scores. The constructive discussion period has significantly strengthened the paper, and the proposed method stands as a solid contribution to the field. Therefore, AC recommends accept.